# A Mutual Information Duality Algorithm for Multi-Agent Specialization

## Abstract

The social behavior change in a heterogeneous population is an essential study of multi-agent learning. The interactions between unique agents not only involves the optimization for single agent learning, agents' behavioral changes also depend on the mutual similarity and dissimilarity between the agents. Our study provides a theoretical derivation of the policies interactions under the formulation of joint policy optimization. We discover that joint policy optimization in a heterogeneous population can affect the population behaviors through mutual information (MI) maximization. In our study, we introduce a minimax formulation of MI (M&M) that optimizes the population behavior under MI minimization against the joint policy optimization. The main findings of our paper show that MI minimization can minimize the behavioral similarity between agents and enable agents to develop individualized policy specialization. Empirically M&M demonstrates a substantial gain in average population performance, diversity, and narrows the performance gap among the agents.

## 1 Introduction

From the success of multi-agent game plays [(OpenAI et al., 2019), (Vinyals et al., 2019)], the field of heterogeneous multi-agent learning is being actively studied in areas such as AI game design (Contributors, 2022), embedded IoTs (Toyama et al., 2021), and the research for future Human-AI interaction. With the unique physique and specialized character attributes in a heterogeneous population, the ability to optimize policies for specific purposes is essential. The difficulty as well as the goal of the heterogeneous population learning research is to search and define a general learning algorithm that optimizes an agent population such that the uniqueness of each agent can be fully utilized within the population. To approach the problem, our research aims to understand the symmetric and asymmetric social behavior changes that occur during the population learning through mutual information formulation.

To study the learning and behavioral change, multi-agent learning has formed two branches of studies. In the simplest form of multi-agent RL, individualized learning is performed to learn separate behavioral policies for each agent. These prior works include Independent Q-Learning (IQL) (Tan, 1993), Policy Space Response Oracle (PSRO) (Lanctot et al., 2017), and AlphaStar (Vinyals et al., 2019). However, the empirical success of these prior approaches have excluded knowledge sharing. Since the training is done independently, one agent's learned experiences do not transfer to another. The individualized behaviors result in high redundancy of re-exploration and low socialized behaviors among the agents. On the contrary, joint policy optimization is proposed as a solution to the listed problems. Joint policy optimization utilizes a single conditioned policy to learn the diverse skillset, and the character attributes of the population via distributed RL optimization. Through population experiences sharing and joint policy optimization, a single conditioned policy network can learn a set of generalized policy skills that are transferable across the different agents. Notable examples include OpenAIFive (OpenAI et al., 2019), HAPPO (Kuba et al., 2021) and NeuPL (Liu et al., 2022) that optimize the multi-agent behaviors under the expected population accumulated rewards.

Through knowledge sharing and joint policy optimization, population learning in interactive games has benefited from the increased learning efficiency, and learning generalization of agents' social behaviors. Our research focuses on the latter, where we analyze the cause of social behavior change.

By analyzing a heterogeneous population with pairwise interactions, we discover that the joint policy optimization of a two-player competitive game converges to *MI maximization* among the agents. Unfortunately, MI maximization optimizes the commonality in a population. We found that in a heterogeneous population where agents are individually unique, MI maximization benefits agents that have characteristics close to the population norm, but severely degrades the performance of unique agents with character attributes dissimilar to the average population. To address the drawback of MI maximization, our research's main contribution is a novel minimax MI (M&M) formulation of population learning that enables individual agents to learn specialization from the dual formation of MI maximization and minimization.

## 2 BACKGROUND AND RELATED WORK

Multi-agent learning is a broad field of study that not only covers the necessary intelligence to achieve individual agent reward maximization, but also the social behavior change when agents interact with other agents. Prior researches in population learning have studied the performance of the agent population in competitive and cooperative environments.

### 2.1 COMPETITIVE BEHAVIORS LEARNING

In a competitive environment, the goal of the multi-agent research is to utilize competition to optimize the performance of an agent population. One approach to perform the learning iteration is individualized learning. Each agent improves its policy through learning a best-response (BR) against the prior iterations of agents. The iterated elimination of dominated strategies optimizes a population of policies under Game Theory. Prior studies such as (Jaderberg et al., 2018), PSRO, PS-TRPO (Gupta et al., 2017), (Vinitsky et al., 2020) and Alphastar utilizes different variants of self-play (Heinrich et al., 2015) to learn competitive Nash Equilibrium behaviors for a population. The different variants addresses the stability (TRPO constraint), robustness (adversarial population) and diversity (leagues of policies pretrained on human data) of individualized learning.

In contrast, a joint policy optimization framework proposes by (Foerster et al., 2016) and (Lowe et al., 2017) optimizes the population with Centralized Learning Decentralized Execution (CLDE). The joint optimization enables common skill transfer across policies. The commonality among the agents can be learned once and the learned behavior can be utilized by different agents of a population. This form of joint policy optimization optimizes the population as an one-body problem with Mean Field Theory (Yang et al., 2018). Under Mean Field Theory, the variations of individual agents can be averaged. The modeling of the population behaviors are reduced from a many-body problem to an one-body problem. Prior works include MADDPG, HAPPO (Kuba et al., 2021), OpenAIFive (OpenAI et al., 2019) and NeuPL (Liu et al., 2022), where OpenAIFive and NeuPL have both further developed efficient graph solvers based on (Shoham & Leyton-Brown, 2008) to optimize the social graphs of the population. The graph solver $F$ optimizes the match pairing of the agents (x,y) with weighted edges $\Sigma^{(x,y)}$. The objective of $F$ commonly optimize the policy learning to be robust against adversarial exploitation or agents with the most similar performance strengths.

### 2.2 COOPERATIVE BEHAVIORS LEARNING

To develop social behaviors of cooperation, prior studies have proposed auxiliary rewards and regularization of mutual information as part of the objective function. This includes OpenAIFive's team spirit reward, (Chenghao et al., 2021)'s Q-value, (Cuervo & Alzate, 2020)'s PPO, and (Mahajan et al., 2019)'s latent variable regularization of MI maximization. While the above studies suggest that learning of cooperative social behaviors require auxiliary modifications with mutual information, (Dobbe et al., 2017) shows an interesting analysis on the distortion rate of the joint policy optimization versus the individualized learning with MI. The study shows that even without auxiliary modification, there is a significant stability difference in distortion rate between the joint optimization and individualized learning. This issue has shown to negatively impact cooperative learning.

## 2.3 A NEW PERSPECTIVE ON EXISTING WORK: FROM POLICY GRADIENT TO MUTUAL INFORMATION

In this section we first show that policy gradient (Sutton et al., 1999) in multi-agent joint optimization can be reduced to MI maximization of an agent population. We derive the learning objective of multi-agent learning as the following.

To model the learning dynamic of a two-players competitive zero-sum game with Player x and Player y, we let $\Pi_\theta(a|s)$ be a population joint policy. Given a game state $s$, players act according to the policy $\Pi_\theta(a|s)$, and let $a^x$ and $a^y$ denote the actions taken by Player x and y. We define $Q(s, a^x, a^y)$, $V(s)$, $Adv(s, a^x, a^y)$ as the centralized learning of Q, value, and advantage functions respectively, where $Adv(s, a^x, a^y) = Q(s, a^x, a^y) - V(s)$. The objective of the policy optimization is to perform maximization of the expected return $J(\theta)$ from episodes of self-play. The joint policy gradient of a single step optimization is then defined by:

$$Maximize\ J(\theta) = \underset{s}{\mathbb{E}}\ \underset{a^x}{\mathbb{E}}\ \underset{a^y}{\mathbb{E}}[Adv(s, a^x, a^y) * (\Pi_\theta^x(a^x|s) \cdot \Pi_\theta^y(a^y|s))] \tag{1}$$

To analyze the behavioral change of the players, we derive Eq(3)'s *gradient* to inspect the direction of *policy change*. We take the derivative with the product rule and apply the log trick (Williams, 1992) to derive the expanded form of the joint gradient:

$$\nabla_\theta J(\theta) = \underset{s}{\mathbb{E}}\ \underset{a^x}{\mathbb{E}}\ \underset{a^y}{\mathbb{E}}[\ Adv(s, a^x, a^y)$$
$$* [\ \Pi_\theta^y(a^y|s)\ \Pi_\theta^x(a^x|s)\ \nabla_\theta log(\Pi_\theta^x(a^x|s))\ +\ \Pi_\theta^x(a^x|s)\ \Pi_\theta^y(a^y|s)\ \nabla_\theta log(\Pi_\theta^y(a^y|s))\ ]\ ] \tag{2}$$

Here we expand the Advantage function to $Q(s, a^x, a^y) - V(s)$. With $\Pi_\theta^x(a^x|s)$ and $\Pi_\theta^y(a^y|s)$ specific to each side of x and y, we assume they are independent. By applying the probability of independence and the logarithmic product rule (A.6), the joint gradient can be defined by:

$$\nabla_\theta J(\theta) = \underset{s}{\mathbb{E}}\ \underset{a^x}{\mathbb{E}}\ \underset{a^y}{\mathbb{E}}[Q(s, a^x, a^y)[[\ \mathbf{\Pi}_\theta^{(\mathbf{x,y})}(\mathbf{a^x, a^y}|\mathbf{s})\ ]\ *\ [\nabla_\theta \mathbf{log}(\mathbf{\Pi}_\theta^{(\mathbf{x,y})}(\mathbf{a^x, a^y}|\mathbf{s}))\ ]\ ]$$
$$-[V(s)[\ \mathbf{\Pi}_\theta^{(\mathbf{x,y})}(\mathbf{a^x, a^y}|\mathbf{s})\ ]\ *\ [\ \nabla_\theta \mathbf{log}(\mathbf{\Pi}_\theta^{\mathbf{x}}(\mathbf{a^x}|\mathbf{s}))\ +\ \nabla_\theta \mathbf{log}(\mathbf{\Pi}_\theta^{\mathbf{y}}(\mathbf{a^y}|\mathbf{s}))]]] \tag{3}$$

The derivation of Eq(3) shows components of mutual information formulation. By taking $T$ steps of joint policy optimization, the integral of Eq(3) removes $\nabla_\theta$ from the equation. The convergence result of the joint policy optimization is a weighted variant of mutual information maximization shown below in Eq(4). This derivation suggests that the long-term behavioral change between two competitive players will converge to *mutual behaviors maximization*.

$$w(x, y) * I(X; Y) = \underset{x}{\mathbb{E}}\ \underset{y}{\mathbb{E}}[[w(x, y) * [\ \mathbf{P_{(X,Y)}}(\mathbf{x, y})\ ]\ *\ [\ \mathbf{log}(\mathbf{P_{(X,Y)}}(\mathbf{x, y}))\ ]]$$
$$-[\ \mathbf{P_{(X,Y)}}(\mathbf{x, y})\ ]\ *\ [\ \mathbf{log}(\mathbf{P_X}(\mathbf{x})) + \mathbf{log}(\mathbf{P_Y}(\mathbf{y}))\ ]] \tag{4}$$

In a heterogeneous population of agents { i, ii,... N}, we extend the two-player game (X,Y) to all pairings of unique agents, the expected joint gradient is then denoted as:

$$\frac{1}{N}\sum_{X=i}^{N}\frac{1}{N}\sum_{Y=i}^{N}\underset{x}{\mathbb{E}}\ \underset{y}{\mathbb{E}}[w(X, Y) * [\mathbb{H}(X) + \mathbb{H}(Y)] - [\mathbb{H}(X, Y)]] \tag{5}$$

The key finding from Eq(3) to Eq(5) is that through the accumulated *gradient* update, the maximization of $J(\theta)$ results in MI maximization in the behavior of a heterogeneous population. We find that the MI maximization formulation can impose a constraint on a heterogeneous population. Particularly agents with character attributes dissimilar to the average population cannot realize their own competitive advantage when their behavioral policies are subject to MI maximization constraint with the average population. An iterative BR variant of joint policy optimization is more difficult to derive the population-level behavior change during the joint policy optimization. For the BR variant, Eq(5) approximates only the population-level behavioral change as both sides of the policy response in a symmetric two-players game are near convergence. i.e. the joint policies utilized by the two players are approximately the same.

## 3 M&M: MiniMax Mutual Information Specialization

We introduce M&M as a MI minimax population learning that aims to address the drawback of joint policy optimization. In particular, we propose a *minimization* of MI to optimize a disjoint set of policies against the MI maximized joint policy optimization. This process enables individual agents to specialize behavioral policies relative to the average population. We denote the *Generalists* as the joint policy that maximizes MI, and can be utilized on the general population. We define *Specialists* as a disjoint set of policies that minimize MI against the Generalists. Specialists' behavioral policies deviate from the population norm through learning behavioral policies that align with agent's innate attributes.

### 3.1 Problem Formulation

We consider a non-Markov two-player zero-sum Game (X,Y) with a heterogeneous population of size $N$ {i, ii... N }. The game is defined by $(O, S, A, R)$ where $O$ is the observation space, $S :$ $O \times O$ is the joint observation of the two players as a fully observable game. The discounted return for each player is defined as $R_t = \sum_{\tau=t}^{\infty} \gamma^{\tau} r_{\tau}$ where $\gamma$ is the discount factor $\gamma \in [0, 1)$. We denote the *Generalists* as $\{\Pi_{\theta^*}^g\}_{g=i}^N$, as an $\epsilon-$Nash Equilibrium (NE) policy solution reached by the joint policy optimization after $T$ iterations of population self-play optimization. Here every player receives an expected payoff within $\epsilon$ of NE as a numerical relaxation of NE. We define and initialize the Specialists as a set of disjoint policies from replication of the Generalists $\{\pi_{\psi_0^k}\}_{k=i}^N \leftarrow$ $\{\Pi_{\theta^*}^g\}_{g=i}^N$, where $\{\pi_{\psi^k}\}_{k=i}^N$ denotes a set of $N$ disjoint policies parameterized by $\psi$ on the 0-th training iteration.

### 3.2 Conditional Mutual Information Minimization

For each set of games, we formulate M&M as a one-vs-all BR policy optimization of a Specialist against the population of the Generalists in a zero-sum two-players game. Given a game state $s$, players act according to their own policy. Let $a^k$ and $a^g$ denote the actions taken by a Specialist $k$ and a Generalist $g$ in a game. We define $Q(s, a^k, a^g)$ and $V(s)$ as the Q and value functions individually for each Specialist. The objective of the policy optimization is to maximize the expected return $J(\psi^k)$ while minimizing MI with the gradient. In each optimization step we fix the parameter of the Generalists and treat the joint policy as a part of the fixed $\epsilon-$NE solution. We then define the effects of $a^g$ as a part of the environment and allow the disjoint policies to condition on the MI maximization joint policy, and perform MI minimization. Formally, we define the conditioned policy gradient for a Specialist $k$ as:

$$Maximize \ J(\psi^k) = \frac{1}{N} \sum_{g=i}^N \mathop{\mathbb{E}}_{(s,a^g)} \mathop{\mathbb{E}}_{a^k} [[ \ Q(s, a^k, a^g) * \pi_{\psi^k}(a^k|s) \ ]$$
$$-[ \ V(s|\Pi_{\theta^*}^g) * \pi_{\psi^k}(a^k|s) \ ]] \tag{6}$$

By treating $\Pi_{\theta^*}^g(a^g|s)$ as the average population's fixed $\epsilon-$NE response, a Specialist $k$'s policy gradient optimization can be simplified to learn a BR against a stationary environment. The derivative of the gradient w.r.t. $\psi^k$ can be defined as:

$$\nabla_{\psi^k} J(\psi^k) = \frac{1}{N} \sum_{g=i}^N \mathop{\mathbb{E}}_{a^k} \mathop{\mathbb{E}}_{(s,a^g)} [Q(s, a^k, a^g) * \pi_{\psi^k}(a^k|s) \ \nabla_{\psi^k} log(\pi_{\psi^k}(a^k|s))$$
$$-V(s|\Pi_{\theta^*}^g) * \pi_{\psi^k}(a^k|s) \nabla_{\psi^k} log(\pi_{\psi^k}(a^k|s))] \tag{7}$$

$$-w(X,Y) * I(X;Y) \equiv -w(X,Y) * [\mathbb{H}(X) - \mathbb{H}(X|Y)] \tag{8}$$

By taking $T$ steps of $J(\psi^i)$ maximization, the integral of Eq(7) maximizes the *negative* of a weighted conditional MI w.r.t. the Generalists denote in Eq(8). Alternatively, the optimization is equivalent to

optimize $\pi_{\psi^k}$ to minimizes its mutual policy response with respect to $\Pi_{\theta^*}^g$ by weighting the individual agent's Q-value gain against the value-function estimation conditioned on the MI maximizing Generalists population.

$$\frac{1}{N}\sum_{k=i}^{N}\frac{1}{N}\sum_{g=i}^{N}-w(X_k;Y_g)*I(X_k;Y_g) \equiv \frac{1}{N}\sum_{k=i}^{N}\frac{1}{N}\sum_{g=i}^{N}-w(X_k;Y_g)[\mathbb{H}(X_k)+\mathbb{H}(X_k|Y_g)] \quad (9)$$

The overall expected social behavior change over time when averaging the disjoint policies' gradients is shown in Eq(9). $w(X_k;Y_g)$ denotes the weighting of $Q(s,a^k,a^g) - V(s|\Pi_{\theta^*}^g)$, and the full formulation shows a minimization effect of MI between every Specialist to the Generalists' joint policy optimization. Utilizing the dual properties of mutual information, M&M optimizes the Specialists population as individual agents that each specializes according to an agent's innate attributes. The benefit of specialization over joint policy optimization is that rather than relying on common skill transfers for policy improvement, a Specialist learns a unique policy that best aligns with its own characteristics. This makes the population social interactions more diverse and enhances the performance of agents with out-of-distribution character attributes. We show an algorithmic implementation of M&M policies optimization in (A.7)

### 3.3 Social Graph Optimization

In our population learning of two-players competitive game, we have implemented a graph solver to both optimize and analyze the social interactions between vertices of agents' policies. Let $F$ be defined as a graph solver, and each agent's policy denotes a graph vertex. The directional weighted edge between two vertices defines the interaction between two vertices as a two-players game. The direction of the edge denotes the relationship between the pairs of vertices as dominance or dominated, and the weight of each edge denotes the probability of the competitive outcome for vertex $k$ as $U^k$. The initialization of all the edges are set to 0.5 as a 50% chance of winning or losing for each pair of vertices. After each batch of finished games, the weights and directions of the edges involved are updated according to the game outcomes. Based on the weights between the vertices, $F$ updates the match distribution of vertex $k$ by defining $\Sigma_k$ as the probability distribution of vertex $k$ forms an interactive game to other vertices. We define the optimization objective of $F$ to prioritize the sampling of matchmaking based on the exploitability of vertex $k$ relative to other policies. Due to the probabilistic sampling of concurrent game interactions, the sampled adversary opponents form a set of mixed strategies we denote as $\sigma$.

## 4 Experiments

Empirically we evaluate the specialization of agents on the Mobile Fighting Game: Naruto Mobile. We compare and contrast the Generalists and Specialists population on advantage analysis, social graph connectedness, behavioral diversity, and agent performance. We discuss the advantage and social graph for each agent to demonstrate how the social behavior change can be largely influenced by the methods of policies optimization. Additionally, our analysis on behavioral diversity and agent performance show that M&M optimizes agents' policies for specialization. By reducing MI, agent population can benefit from behavioral diversity and performance gain.

### 4.1 Intro to Naruto Mobile Game Mechanics

**Game mechanics**: Naruto Mobile is a real-time 2D mobile Fighting Game of 1 vs 1 gameplay. The game includes a pool of over 300 unique characters. Each character is designed with a unique set of character attributes such as attack speed, range and movement variations. Additionally, each character has skills of different cooldown, status effects and duration. The large population of characters and real time competitive interactions provides a testbed for research on a heterogeneous population. Before the start of a match, both sides of players start by selecting their characters, auxiliary skill scroll and a pet summon. The key to winning a match is to reduce the opponent's health point (HP) to zero, or having a higher HP at the end of a 60 second match timer.

## 4.2 AGENT SPECIALIZATION

We record the advantage of the Generalists' and the Specialists' population against their prior iteration of population policies. The recorded advantage vectors of agent $k$ comprise the expected advantage of agent $k$ against its opponents over the duration of evaluation. The recorded advantage vectors of an agent are projected on the unit circles to visualize the vectors' magnitude and the vectors' direction components. The radial plots are measures in the population's mean-deviation with the center of the circle representing zero advantage over the average population, and the radial edges represent the maximal deviation gain against a specific radial opponent.

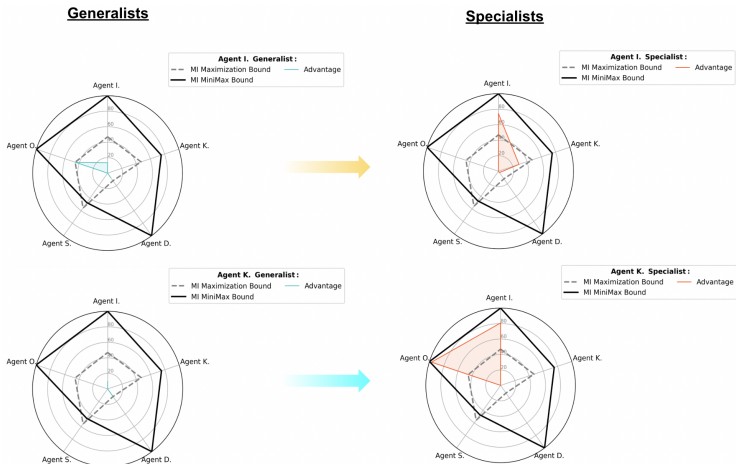

Figure 1: **Agent's Advantage And Specializations:** From MI maximization optimization, both Generalists Agent I. and Agent K. show low deviations of advantage relative to the general population. This shows MI maximization optimizes the marginalized advantage of the population. In contrast, the Specialists Agent I. and Agent K. have demonstrated strong deviation from the population norm. With MI minimization being the optimization objective, Specialists learn to gain an additional competitive edge by specializing agents' unique agent-attributes.

From before to after agents' specialization, Figure 1 shows Agent I. has an increased magnitude of agents' advantage vectors. This implies that under M&M specialization, the MI between Agent I. and the general population is minimized. Similarly, Agent K. is also able to minimize its MI to achieve its own specialization. While both agents are able to achieve specialization, the direction components are different for the two agents. Agent I. is specialized against prior iteration of itself and Agent K., and Agent K. is specialized against prior iteration of Agent I. and Agent O.. The unique directions of each agent's specialization shows that M&M specialization not only minimizes MI of the population, each agent is also optimized for its own agent-attributes.

Moreover, the Generalists and Specialists populations show different aggregated bounds. With the aggregate of Generalists as the dotted line region and the Specialists population as the solid line region, the radial plot shows that the Generalists dotted bound is restricted close to the center. With the center representing the average population, the closer the region to the center, the more generic the agent population behaves. In the extreme case of conformity behavior convergence, MI maximization can reduce the region and eliminate the diversity in the population. On the other hand, Specialists show individual differences relative to the population norm. The individual specialization creates a collective diversity for the population.

## 4.3 SOCIAL GRAPHS AND INTERACTION STABILITY

With NeuPL (Liu et al., 2022) and (Shoham & Leyton-Brown, 2008) graph solver, the different social graphs of the Generalists and Specialists are plotted on their corresponding radial charts. In a social graph, the labeled agent is the center node of the graph and is connected with edges $\Sigma(x, y)$ outward to other agent nodes. $\Sigma(x, y)$ are normalized with population mean-deviation. The MI Maximized Social Graph is shown on the left in blue, and M&M's Social Graph on the right in red.

The social graphs display the relative frequency agents interact with each other and the deviation from the mean. The larger an edge value is, the more frequent the agent-nodes interact.

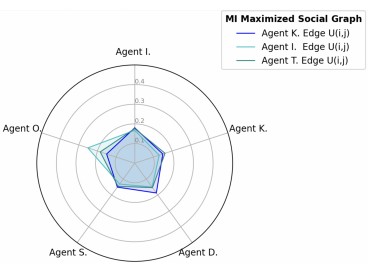 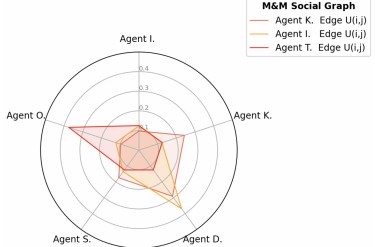

Figure 2: **Social Graphs of Generalists and Specialists:** In MI Maximization Social Graph, the uniformity of edge-connection strengths shows each agent approximately interacts with other agents a similar number of times. This result shows each agent receives approximately the same distribution of experiences. In contrast, the social graphs of the Specialists gradually shift away from the symmetry. This allows each agent to interact with more relevant agents, and acquire more individualized information distribution. The differences in social graphs directly influence the stability of population learning and agents' experiences distribution. The more disjoint the experience distributions are, the sparser some interactions become.

MI Maximized Social Graph shown in Figure 2 illustrates that agents are connected with approximately equal connection strength in each edge. This indicates a densely connected social graph. This approximately translate to uniform distribution of $\{P_{X,Y}(x,y)\}_{x,y=i}^{N}$ with high stability in $-\mathbb{H}(X,Y)$. In contrast, M&M's Social Graph shows few strongly connected edges, such as Agent T. to Agent O., and sparse connections with other agents. With sparsity in the social graphs, each agent is to receive a different distribution of experiences.

### 4.4    FROM MONOTONE BEHAVIOR TO DIVERSE INTERACTIONS:

To analyze the agents' behavior, we introduced several metrics to track the *timing* of agents' skills. ForcingMoves are metrics that measure agent's attack initiation prior to the opponent's attacks. CounterMoves measure skills used to interrupt the opponent *during* their skill casting. CloneSubstitute measures the frequency that an agent avoids or escapes from the opponent's attack. The metrics are displayed via mean-deviation frequency to account for the different skill cooldown among the agents. We collect a batch of 100 games for each agent-type to look at how similar or different the learned strategies are.

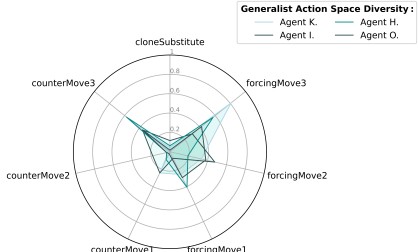 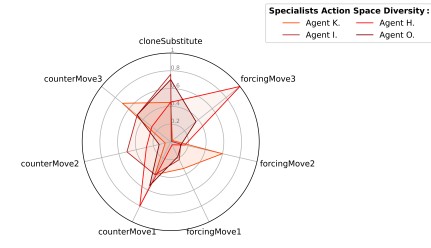

Figure 3: **Timing and Strategy:** Comparing the behavioral difference between the Generalists and the Specialists, there is a clear difference in how Generalists and Specialists respond. For the Generalists, the mean-deviation shows similar change despite playing four uniquely different agents. Except for the cooldown constraint, the actions of the four Generalists are largely the same response. This shows the drawback of MI maximization optimization. On the contrary, the Specialists are incentivized to formulate creative plays based on their individual agent-attributes. M&M creates a wide variety of strategy timings and competitive behaviors.

In Figure 3, the stacked Generalists and Specialists action timing reveals the similarity and differences in different population's strategy timing. Specialists learn to time their skills differently when embodied in different agents. Each agent adapts its behavior such that its current strategy is optimal relative to the general population. In contrast, the Generalists population shows low variations in strategies. With the similar action timing, the Generalists have developed social behaviors that *conform* with other agents. While under MI maximization, the optimized generic behavior works across the different agents, but the diverse Specialists behaviors better demonstrate the essence of general intelligence for adaptive learning and creativeness.

### 4.5  PERFORMANCE COMPARISON:

The $\epsilon$ - Nash of the Generalists agents is iteratively optimized with 13 rounds of joint policy optimization via NeuPL's graph solver (Heatmap shown in Appendix A.4). Here we display the top 8 performing Generalists relative to the M&M optimized Specialists. The displayed competitive performance are the weighted win rates of the 8 agents versus the $\epsilon$ - Nash of the Generalists population.

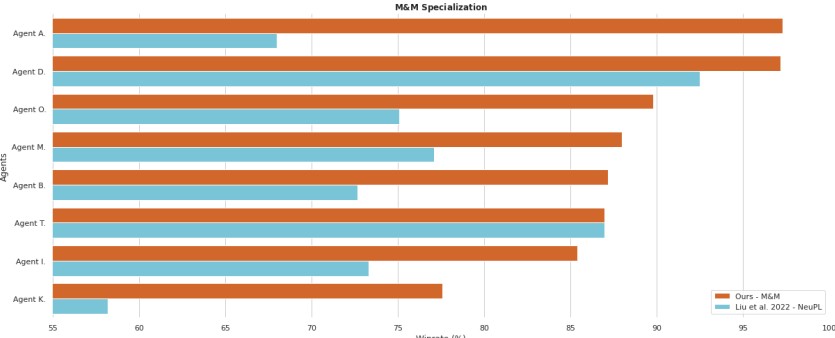

Figure 4: **Performance Re-Balance with M&M Specialization:** In experiment 4.5 we look at how agents' performances can be better balanced with M&M specialization. We first observe that with joint policy optimization, the top performing NeuPL agents converge to win rates of 58.2 ̃92.5% in deep blue with high performance inconsistency among the agents. The variation of performance shows that the joint policy may converge towards a set of behavioral strategies that benefits some agents more than the others. Particularly, MI maximized joint policy benefits the average population by adapting mutually similar joint policy, but for agents such as Agent A. and Agent K. their character attributes may fall outside of the population norm. We see that with M&M, Agent A. and K. demonstrate a substantial performance gain and re-balanced the population performance gap (77.6 ̃97.3%). Our empirical result shows that minimizing MI can enable agents to search for uncommon strategies that outperform the MI maximized counterpart.

Aside from the performance gain from agent specialization, Figure 4 also shows an interesting observation that optimizing for population's commonality (MI maximization) does not translate to a more balanced population. With joint policy optimization, the population converges with high performance differences among the agents. The standard deviation of MI maximized optimization is 9.95, while it is reduced to 5.99 after M&M's specialization. With approximately 30 % reduction in population performance deviation, it shows that equity of a population (performance gap) can be improved when individual agents specialize.

### 4.6  SAMPLE EFFICIENCY:

In our last experiment we look at the sample efficiency of the different optimizations. The joint policy optimization of the Generalists are performed on a set of 16 agents under MI maximization for 9 rounds. We then compare the relative performance gain to sample required with joint policy optimization and M&M for 2 additional rounds of optimization. The experiment measures the sample complexity needed for the two methods to reach a similar level of relative performance prior to specialization.

| Model Name | Total Training Steps | 9th Round Generalist | 11th Round Generalist | Agent K. Specialist | Agent M. Specialist |
|---|---|---|---|---|---|
| 9th Round Generalist | 720,000 | - | 33.9% | 21.1% | 6.4% |
| 11th Round Generalist | 890,000 | 66.1% | - | 44.6% | 19.1% |
| Agent K. Specialist | 850,000 | 78.9% | 55.5% | - | 36.2% |
| Agent M. Specialist | 850,000 | 93.6% | 80.9% | 63.7% | - |

Figure 5: **Training Efficiency and Competitiveness**: With 170k additional learning steps, the 11th round of Generalist can efficiently increase the average performance of the 16 agents by 16.1% (50% ~66.1%). With M&M specialization, Agent K. and Agent M. can reach the same relative win rate against the general population as the 9th round of population with an additional 130k learning steps. The experiment indicates that when optimize for MI maximization, experience sharing allows joint policy optimization to efficiently optimize the expected behaviors of the whole population on commonality. Since MM optimization no longer optimize with MI maximization as an objective, each experience sample is unique to each agent and would create separate learning distribution for each agent. To collect each separated learning experiences, additional computation is required.

In Figure 5, joint policy optimization demonstrates that it can use less learning steps per agent to optimize a population of agents. In contrast, due to the diversified exploration M&M requires additional samples to explore the set of MI minimized agents interactions. The main takeaway from this experiment is that it would be more learning efficient to first learn the common skills under joint policy optimization. Sequentially, after Generalists' convergence, M&M specialization can further benefit each agent with additional samples of training that diversify the learning experiences and enables specialization.

## 5 CONCLUSION

In our paper, we have derived two formulations of MI from multi-agent policy gradient. One brings a new perspective on the existing work of joint policy optimization, the other explores the potential of conditional MI minimization. The two learning representations under the dual formulations of MI measure the long-term effects of a population's social behavior change. With the joint policy optimization, the MI maximization of gradients significantly increased the sample efficiency of population learning by learning a set of transferable skills across the agent policies. Through maximizing MI as policy's gradient, the joint policy optimization optimizes the heterogeneous population toward a set of mutually similar behaviors among the agents.

In contrast, our proposed M&M population learning aims to minimize the MI between the disjoint policies and the MI maximized joint policy agent population. Our conditional MI minimization population learning enables each agent to define an unique policy that best aligns with the agent's own character attributes. Without the MI maximization constraint, the learned policy does not need to be transferable to other agents. In a heterogeneous population the specialization of behavioral policies are especially beneficial to agents that have dissimilar character attributes relative to the average population. As a result, M&M can optimize a diverse population of Specialists that better balance the performance inconsistency within a population. We believe that through studying the gradient formulation of population policy gradient can help to bring better understanding on the social behavior change that occurs within a population.

**Future Work:** The current limitation of our method is on the additional compute resource for learning agent's individualized specialization. Due to the process of specialization being specific to each agent's character attributes relative to the joint policy, sample efficiency improvement on joint and disjoint policies optimization can both benefit the training needed for social behavior study. We thank the reviewers for suggestions and their valuable constructive feedback on our study.

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

# A APPENDIX

## A.1 MOBILE DEEP LEARNING ARCHITECTURE

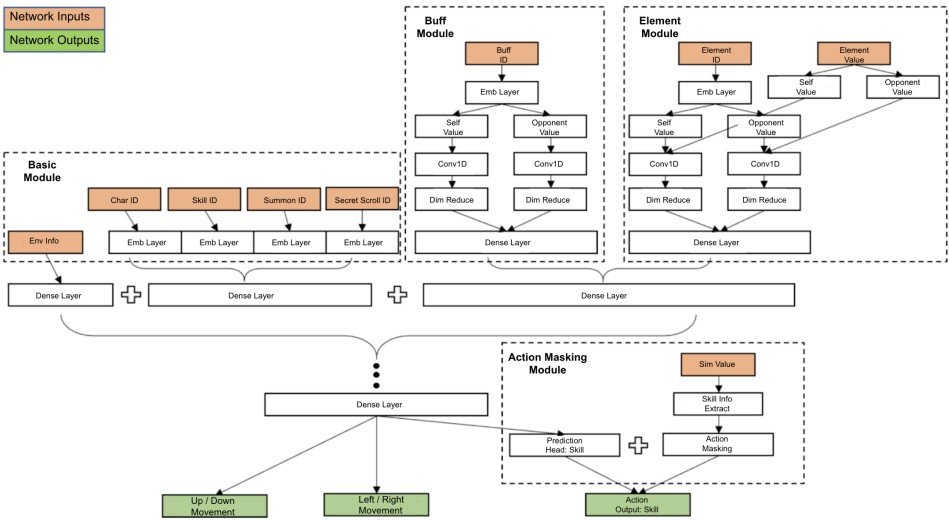

Figure 6: **DRL architecture for mobile devices:** We build our neural network deployable to mobile devices. We first pass inputs to an embedding layer to reduce the dimension and also use Conv1D as opposed to the Dense layer to reduce compute load. Our input to output is separated into four modules.

1. Basic Module - for game env stat & characters specific information.
2. Buff Module - buff and debuff info during gameplay.
3. Element Module - characters equipment of weapon IDs, armor, etc.
4. Action Masking Module - gives the model the information on whether each action is currently available.

The separation of modules in Figure 6 allows us to have sparse connectivity in the model. The compute load is reduced compared to fully connected layers. This allowed us to inference our models locally on mobile devices.

**MDP:** The model receives from the Basic Module to identify the characters and equipped Summons and Scrolls. Additionally, characters position, movement and skills' buff & element information are received from Buff and Element Module. The above are the input states $s_t$ to our model. Our model predicts an action output $a_t$ to control the 2D movement of the agent, and the available attack and skills. The reward can be customized, but in our standard mechanics it is based on the (weighting) of an agent's own HP(10), opponent's HP(10), the result of the battle(10), combo(5), and mana(5). With the transition of the action, new state $s_{t+1}$ are given to the model for the next iteration of MDP.

### A.1.1 HYPERPARAMETERS AND HARDWARE USED

Hyperparameters

- PPO: 0.1
- n-step: 100 frames
- Reward discount factor: 0.995
- Learning rate: 1e-4

Hardware Used

- CPUS: 5,300
- GPUS: 0
- max Batch size: 5120
- Average Compute Time:
    - ≈ 180 Hrs (Generalist Training)

## A.2 Heterogeneous Action Space Representation Learning:

With a large collection of distinct agents, the learning problem we face is their unique action sets and their relative similarity. Inspired by word embedding (Mikolov et al., 2013), we approach the problem with a structure of information bottleneck to learn the representation of the action sets as a dense representation of a vector. We define our information bottleneck via the structure of a neural net $\Pi(a|S)$, where $S$ is the learn-able state space that holds the contextual information of the involved agents $(x, y)$, action sets $(a^x, a^y)$ and the joint observations. The goal of the model structure is to compress the large contextual information $S$ to learn a joint policy $\Pi$ that captures the actions' similarity in a specific interaction context.

## A.3 Independent Causality Between Agents' Interactions:

VM3-AC (Kim et al., 2020) has pointed out the possible causality dependency of multi-agent interactions. When player $i$ decides its action $a_t^x$ it may depend on player $j$'s action $a_t^y$. The obvious problem of reasoning is the circular causality of the two players, $a_t^x$ depends on $a_t^y$ and $a_t^y$ depends on $a_t^x$. To resolve this issue, (Kim et al., 2020) has introduced a random variable to disassociate the loop dependency via variational lower bound.

We define our agents' causality graph without the Markov Assumption on agent's action. By removing the Markov *Assumption*, we can use opponent's past actions $a_{0:t-1}^{-\pi}$ to predict opponent's current move $a_t^{-\pi}$. This breaks the loop and assume the casual graph of players' actions are independent at time $t$. This gives the agents temporal consistency of Forward Induction (Battigalli & Siniscalchi, 2002) of Game Theory for self-play. The reached Nash Equilibrium under Forward Induction is self-consistent that the rational opponents in the past will continue to choose rationally in the future.

## A.4 Generalists Self-play Algorithm Construction:

Our neural population learning for joint policy optimization is performed under the multi-agent social graph of NeuPL(Liu et al., 2022). The nodes are the different generations of agents, and are connected by the weighted edges of $\{\Sigma^{(x,y)}\}^{NxN}$. NeuPL provides a population self-play framework that not only competes the current population $\Pi_{\theta_\tau}^N$ with combinations of distinct agents, but the weighted edge prioritization also extend to different *generations* of $\epsilon$-Nash population $\Pi_{\theta_{0:\tau-1}}^N$. Each population is represented as a conditional joint policy that learns a set of BR strategies against all previous generations of multi-agents mixed-strategies.

| Training Round | Behavior Tree | Round 1 | Round 2 | Round 3 | Round 4 | Round 5 | Round 6 | Round 7 | Round 8 | Round 9 | Round 10 | Round 11 | Round 12 | Round 13 |
|---|---|---|---|---|---|---|---|---|---|---|---|---|---|---|
| Behavior Tree | - | 0.94% | 0.21% | 0.14% | 0.22% | 0.29% | 0.23% | 0.12% | 0.12% | 0.08% | 0.07% | 0.09% | 0.06% | 0.09% |
| Round 1 | 99.06% | - | 2.56% | 1.93% | 2.12% | 2.06% | 2.57% | 2.68% | 2.01% | 2.23% | 2.41% | 2.46% | 2.01% | 1.94% |
| Round 2 | 99.79% | 97.44% | - | 4.48% | 4.66% | 5.75% | 7.08% | 6.24% | 5.44% | 5.64% | 6.13% | 6.71% | 5.11% | 4.91% |
| Round 3 | 99.86% | 98.07% | 95.52% | - | 5.91% | 6.90% | 8.86% | 9.72% | 9.18% | 10.58% | 9.95% | 10.78% | 8.29% | 8.89% |
| Round 4 | 99.78% | 97.88% | 95.34% | 94.09% | - | 14.29% | 16.56% | 21.59% | 20.24% | 20.82% | 17.84% | 18.25% | 15.69% | 15.69% |
| Round 5 | 99.71% | 97.94% | 94.25% | 93.10% | 85.71% | - | 18.70% | 21.92% | 20.82% | 21.43% | 21.56% | 21.77% | 19.78% | 19.33% |
| Round 6 | 99.77% | 97.43% | 92.92% | 91.14% | 83.44% | 81.30% | - | 17.50% | 17.33% | 18.12% | 20.03% | 20.05% | 20.05% | 19.88% |
| Round 7 | 99.88% | 97.32% | 93.76% | 90.28% | 78.41% | 78.08% | 82.50% | - | 25.02% | 25.69% | 26.21% | 26.16% | 25.68% | 24.82% |
| Round 8 | 99.88% | 97.99% | 94.56% | 90.82% | 79.76% | 79.18% | 82.67% | 74.98% | - | 33.16% | 32.85% | 31.30% | 31.24% | 31.81% |
| Round 9 | 99.92% | 97.77% | 94.36% | 89.42% | 79.18% | 78.57% | 81.88% | 74.31% | 66.84% | - | 33.87% | 33.96% | 31.71% | 33.28% |
| Round 10 | 99.91% | 97.54% | 93.29% | 89.22% | 81.75% | 78.23% | 79.95% | 73.84% | 68.70% | 66.04% | - | 41% | 32.12% | 33.89% |
| Round 11 | 99.93% | 97.59% | 93.87% | 90.05% | 82.16% | 78.44% | 79.97% | 73.79% | 67.15% | 66.13% | 59% | - | 36.10% | 36.71% |
| Round 12 | 99.94% | 97.99% | 94.89% | 91.71% | 84.31% | 80.22% | 79.95% | 74.32% | 68.76% | 68.29% | 67.88% | 63.90% | - | 42.39% |
| Round 13 | 99.91% | 98.06% | 95.09% | 91.11% | 84.31% | 80.67% | 80.12% | 75.18% | 68.19% | 66.72% | 66.11% | 63.29% | 57.61% | - |

Figure 7: The heatmap shows the evaluation matches across the different Generalist iterations $\Pi_{\theta_t}^N$. The ablation evaluation shows joint policy optimization roughly diminished as the training approaches the 11th 13th iteration. In particular, the 13th iteration only has a 57.6 % winrate against the 12th iteration of Generalist, which is close to the Nash equilibrium of 50 %.

In Figure 7, we evaluate all rounds of the Generalist policies against each other. The heatmap shows monotonic convergences of the Generalist population. At the 13th iteration, the Generalists population has converged as $\varepsilon-$ Nash Equilibrium where $\varepsilon \approx 7.6\%$. Further training may minimally increase the performance, but worsen the strategy diversity.

## A.5 Correlation of Diversity and Performance

Since Agent T. is the only agent without performance gain after M&M specialization, we compare its Advantage vector change, performance chart with Agent H.'s.

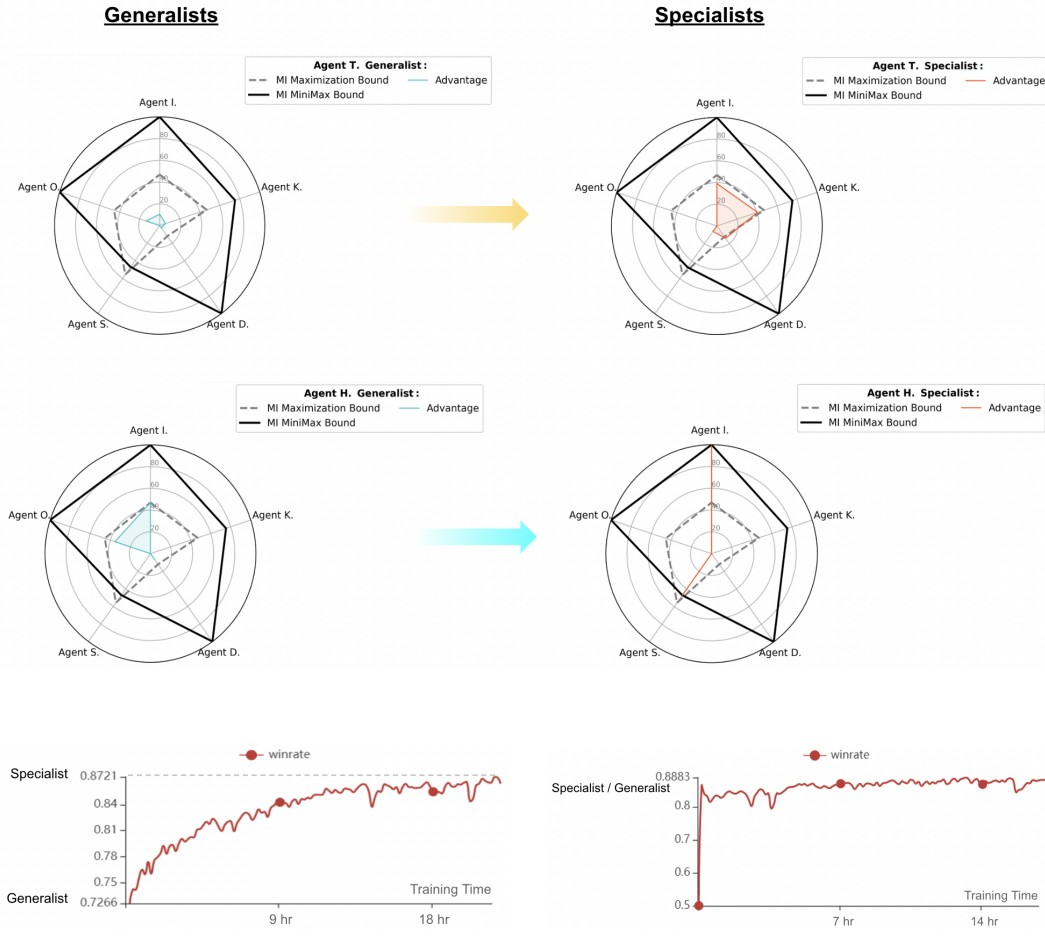

Figure 8: **The performance metric of Agent H. and T. Specialization:** Agent H. shows an initial performance of 72.7 % performance relative to the other 15 Generalists population. With M&M's specialization, Agent H. shows a substantial increase in competitive performance of 87.2 %. On the other hand, Agent T. shows an initial performance of 87 % performance relative to the other 15 Generalists population. With M&M's specialization, Agent T.'s performance does not deviate from the initialization.

From Figure 8, shows the positive correlation of strategy diversity and character performance still holds. With Agent H., the radial plot shows a large deviation when the strategy is being converted from Generalist to Specialist. This correlates with the notable increase in agent performance on the left chart of Figure 8. On the other hand, Agent T. shows a more muted deviation within the Generalist bound. The lower strategy deviation is also reflected in Agent T.'s flat performance on the right chart of Figure 8. The lack of specialization for Agent T. may be due to the Generalists' converged strategy is already close to the optimal strategy for Agent T. This may imply that Agent T. has the most centered agent attributes in relation to the population. As the Generalist policy

converges towards the population mean, the converged strategy can become the optimal strategy for Agent T.

### A.6 INTEGRATION:

Let $\Pi_\theta^x(a^x|s) = x$, $\Pi_\theta^y(a^y|s) = y$, and x, y > 0 then the change of variable approach (mickep , https://math.stackexchange.com/users/97236/mickep) (user65203):

$$\text{Let log(x)} = \int_{t=1}^{x} \frac{\partial t}{t} \tag{10}$$

$$log(xy) = \int_{t=1}^{xy} \frac{\partial t}{t} \ = \int_{1}^{x} \frac{\partial t}{t} + \int_{t=1}^{xy} \frac{\partial t}{t} - \int_{t=1}^{x} \frac{\partial t}{t} \ = \int_{t=1}^{x} \frac{\partial t}{t} + \int_{t=x}^{xy} \frac{\partial t}{t}$$

$$\text{Let } u = \frac{t}{x}, \text{such that} \frac{\partial t}{t} = \frac{\partial u}{u}$$

$$= \int_{t=1}^{x} \frac{\partial t}{t} + \int_{u=1}^{y} \frac{\partial u}{u} \tag{11}$$

This gives us $log(Pi_\theta^{(x}(a^x|s)Pi_\theta^{(y}(a^y|s)) = log(\Pi_\theta^{(x}(a^x|s)\Pi_\theta^{(y}(a^y|s))$. With the probability of independence assumption, $Q(a^x, a^y, s)$ function has the joint probability of $\nabla_\theta log(\Pi_\theta^{(x,y)}(a^x, a^y|s))$ in Eq(3).

### A.7 ALGORITHM PSEUDOCODE

In this section we breakdown M&M population learning into pseudocode, where the main components of the Specialists optimization are the neural population learning (NPL), $\epsilon-$NE of Generalists policy ($\Pi_{\theta^*}^n$), Specialists policies ($\{\pi_{\psi_0^k}\}_{k=i}^N$) and graph solver $F$. Let NPL be the concurrent optimization of two-players game between two agents' policies. For every episode NPL simulates the match of $(\pi, \sigma, \Pi)$, where $\pi$ is matched against an agent member of $\Pi$ and the population of $\Pi$ collectively plays a mixed strategy $\sigma$. We collect the policies interaction trajectories, $\mathbb{T}$, into a replay buffer for policy gradient optimization.

---

**Algorithm 1** Neural Population Learning By RL - NPL($\pi, \sigma, \Pi$)

---

$replayBuffer \leftarrow \{ \}$                    // Initialize trajectory replay buffer
**for** $Episode \in 1, ..., t$ **do**
    Start game engine with policies $(\pi, \Pi)$.
    Store trajectories $\mathbb{T}$ from policies interactions $(\pi, \sigma, \Pi)$ into $replayBuffer$.
    $replayBuffer \leftarrow replayBuffer \bigcup \mathbb{T}$
**end for**
**return** $replayBuffer$

---

For a given heterogeneous population $\{i, ii, ...N\}$ and an $\epsilon-$NE Generalists population $\Pi_{\theta^*}^N$, M&M optimizes $\{\pi_{\psi_0^k}\}_{k=i}^N$ to minimize MI of the individual agent policy against the Generalists population $\Pi_{\theta^*}^N$. Each one-vs-all matchmaking is sampled based on the priority given by the graph solver $F$. After a batch of episodes of NPL, we optimize $\pi_{\psi_\tau^k}$ with PPO optimization. Additionally, we define Eval() to compute the aggregate outcomes of NPL.

---

**Algorithm 2** M&M Multi-Agent Specialization Pseudocode

---

**Input**:

 Population = {i,ii,...N}        // Heterogeneous population of N distinct agents

 $\{\pi_{\psi_0^k}\}_{k=i}^N,$           // N disjoint policies

 $\Pi_{\theta^*}^N,$              // $\epsilon-$NE Generalists

 $\{\Sigma^k := (\pi_{\psi^k}, \{\Pi_{\theta^*}^g\}_{g=i}^N)\}_{k=i}^N,$   // Agent k's social graph

 $F : \mathbb{R}^{1 \times N} \to \mathbb{R}^{1 \times N}.$      // Graph solver - NeuPL (Chen et al. 2022)

**Parameter**: $\Pi_{\theta^*}^N, \{\pi_{\psi_0^k}\}_{k=i}^N.$

**Output**: $\{\pi_{\psi_T^k}\}_{k=i}^N, \Pi^N, \{\Sigma^k\}_{k=i}^N$

**Algorithm Starts**:

1:  **for** $n \in Population$ **do**
2:   $\pi_{\psi_0^n} \leftarrow \Pi_{\theta^*}^n$        // initialize N disjoint policies
3:  **end for**
4:  **while** (true do) **do**
5:   **for** $k \in Population$ **do**
6:    $\Pi^{\Sigma^k} \leftarrow \{\Pi^g(a^g|s)\}_{g=i}^N$   // Specialist k's social graph with the Generalists
7:    $NPL(\pi_{\psi_\tau^k}, \sigma_k, \Pi^{\Sigma^k})$    // One-vs-all population learning with $\pi_{\psi_\tau^k}$
8:    $\pi_{\psi_\tau^k} = PPO_{clip}(gradientStep(\pi_{\psi_\tau^k}))$// Optimize policy with PPO optimization
9:    $U^k \leftarrow Eval(\pi_{\psi_\tau^k}, \{\Pi^g\}_{g=i}^N)$  // Eval() computes the aggregate values of
10:                // vertex k's game outcomes.
11:    $\Sigma^k \leftarrow F(U^k)$       // Define k's social graph
12:   **end for**
13:   $\Pi^N = \Pi^N \cup \{\pi_{\psi_{\tau^*}^k}\}_{k=i}^N$   // Adding Specialists to opponent pool
14:              // Iteratively specialize Specialists $\{\pi_{\psi_\tau^k}\}_{k=i}^N.$
15:  **end while**
16:  **return** $\{\pi_{\psi^k}\}_{k=i}^N, \Pi^N, \{\Sigma^k\}_{k=i}^N$

---

With Section 3.3's defined social graph, we use $U^k$ to denote the probabilistic outcome distribution of all pairwise game matches with Specialist $k$. We use $F(U^k)$ to update $k$'s social graph weighted edges to prioritize sampling of adversarial opponents. After all Specialists have converged, we add the population into the opponent pool set. M&M iteratively optimizes the set of Specialized population until population performance convergence.

## A.8  SOFTWARE AND LICENSING

The models are implemented via Tensorflow, TensorflowLite (Abadi et al., 2015), IM-PALA(Espeholt et al., 2018), and Horovod(Sergeev & Balso, 2018). These softwares are all licensed under Apache License 2.0.

