# OpenReview forum: "A Mutual Information Duality Algorithm for Multi-Agent Specialization"
_ICLR.cc/2023/Conference — Submitted to ICLR 2023_

### Official Review · Reviewer_Eoc4 · 2022-10-26

**Confidence:** 3
**Correctness:** 2
**Technical Novelty And Significance:** 2
**Empirical Novelty And Significance:** Not applicable
**Recommendation:** 3

**Clarity, Quality, Novelty And Reproducibility:**

### Clarity:
The paper writing is casual and informal. Major revision is needed.

### Quality/Novelty
The unclear writing makes it hard to judge the paper’s technical quality and novelty.

### Reproducibility:
No code is provided and the presentation of the technical content is unclear. The reviewer thinks the results may be difficult to reproduce.


**Strength And Weaknesses:**

### Strength:
The paper studied social behavior change and agent specialization, which are important problems for multi-agent reinforcement learning.

### Weakness:

The writing of the paper is casual and lacking the rigor needed by a research paper. Many terms and symbols are undefined. Many sentences and paragraphs are unclear. Please see the following for more details:
1. In Eq 1, symbols such as $\theta, a^x, a^y, s, \pi$ and the term ADV are all undefined. Without defining the symbol, the authors left the readers guessing what each symbol meant.
2. In P3, Q, V are undefined.
3. In P.3, what are Generalists and Specialists? Please provide a formal definition.
4. In P.4, what is a graph solver F? Please provide definition and context.
5. In P.5, what is a ‘recorded vector’? Please define and explain.
6. The term social graph is used throughout the paper without definition.

The above list is not exhaustive. There are still a lot of undefined terms.


**Summary Of The Paper:**

This paper studies agent specialization for multi-agent reinforcement learning. The authors proposed a min-max formulation of mutual information. The authors claim that the proposed method enables agents specialization with stable regularization. The proposed approach is evaluated on the newly introduced ‘Naruto Mobile’ which  is a real-time 2D mobile game with two competitive players.

**Summary Of The Review:**

In summary, the reviewer found the paper not ready for publication due to the writing quality.

---

> ### Author Response · Authors · 2022-11-07
> **Writing Formalism Feedback From Reviewer**
>
> We apologize for the our inadequate paper writing, and we agree with the Reviewer's judgement on the formalism of paper writing.
>
> The undefined notation, function and terms as pointed out by the Reviewer will be addressed.
>
> We thank the Reviewer for the feedback.

---

> ### Author Response · Authors · 2022-11-11
> **Paper Revision #1**
>
> We have included many suggestions from your constructive feedback into our revised paper.
> For our first paper revision, we have uploaded a red-line / boxed highlight revision of our paper.
>
>       The red-lines address the specific issues pointed out by the individual reviewers.
>
>       The boxed highlighter highlights portions of paper that are added to better clarify, define the undefined notations, terms, definitions of social graph and more.
>
>
> Here is what we have done on the paper revision:
>
>       1. We added introductory paragraphs to problem formulation, precise notations and functions definitions to improve - writing clarity
>
>       2. A couple of paragraphs (i.e. section 2.3, 3) have been rewritten with clean descriptions of the proposed methodology to improve - writing clarity.
>
>       3. Generalists, Specialists, social graph definitions and descriptions have been added to the paper explicitly.
>
>       4. We moved our pseudocode into the appendix for extra space in the main text.
>
>       5. We added additional details on Experiment 4.5’s analytic discussion.
>
>       6. We have extended Conclusion to summarize the paper.
>
> We are thankful for the invaluable feedback from the reviewers.
> The paper clarity has notably improved because of our discussion.
>
> For the following week, we appreciate your time and would like to seek additional feedback. We will continue to improve the paper quality until the paper is satisfactory to you.
> Thank you!

---

> ### Comment · Reviewer_Eoc4 · 2022-12-11
> **Thanks for the revision**
>
> Thanks to the authors for the revision. The reviewer found the clarity of the paper still needs improvement.

---

### Official Review · Reviewer_BFMo · 2022-10-30

**Confidence:** 3
**Correctness:** 3
**Technical Novelty And Significance:** 3
**Empirical Novelty And Significance:** 3
**Recommendation:** 3

**Clarity, Quality, Novelty And Reproducibility:**

Clarity: Very unclear and confusing

Quality: The quality of the figures can be improved and the explanation of the results can be enhanced by making the key information clear.

Novelty: The considered problem is interesting and novel, but the methodology is confusing to me.

Reproducibility: The author did not provide the source code, making it hard to evaluate the reproducibility.



**Strength And Weaknesses:**

Strength:

The paper provided a new perspective on the existing work of joint policy optimization, The experimental environment selected a novel Naruto game. Naruto Mobile is a real-time 2D mobile Fighting Game of 1 vs 1 gameplay. The game has a pool of over 300 unique characters, and each has different character attributes in attack speed, range and movement variations. Additionally, each character has skills of different cooldown, status effects and duration. The large population of characters and real time competitive interactions create a testbed environment for research on heterogeneous specializations, embedded agents and Game Theory of Nash Equilibria.

Weaknesses:

* The writing of this paper is confusing, and there are many basic mistakes. For example, PSRO is Policy Space Response Oracle, not Policy Search Response Oracle.

* The objective function in Eq.(1) is confusing. As this is a competitive game, how is the advantage function $Adv(s,a^x,a^y)$ computed?

* In eq(2), how could the policy gradient of two competitive players be computed together?

* It’s not clear to me how mutual information helps in multi-agent specialization.

* The definition of $\epsilon$-nash is not given.

* This work would benefit from testing on more tasks.


**Summary Of The Paper:**

This paper proposed a minimax formulation of MI (M&M) that enables agents specialization with stable regularization. In this paper, authors have derived two formulations of MI from multi-agent policy gradients. One brings a new perspective on the existing work of joint policy optimization, the other explores the potential of conditional MI minimization. Then authors empirically evaluated M&M against the prior SOTA MARL framework, and analyzed the social behavior change in performance, diversity, and the stability of their social graphs.


**Summary Of The Review:**

The paper provided a new perspective on joint policy optimization to enable agent specialization. The considered problem is interesting. But the reviewer found the present methodology is hard to follow and the experiments can be enriched by testing on more scenarios.

---

> ### Author Response · Authors · 2022-11-07
> **First Thread: Paper Discussion With Reviewer On Competitive Game Implementation**
>
> Q1: The objective function in Eq.(1) is confusing. As this is a competitive game, how is the advantage function $Adv(s,a^x,a^y)$ computed?
>
>       A1: We believe in most competitive games computing $Adv(s,a^x,a^y)$ should be possible, since the advantage function is used for policy gradient post-inference. This means after a competitive match, the joint policy will use the advantage function to evaluate information that has already happened and learn from the interactions. In terms of technical implementation,  s, a^x, a^y are stored as a tuple in the joint policy’s replay buffer during the match. After the match,  s, a^x, a^y are sampled to compute policy gradient for learning.
>
> Q2: In eq(2), how could the policy gradient of two competitive players be computed together?
>
>       A2: For a two players competitive game with players x and y, let x’s payoff matrix = A and y’s payoff matrix = B. When the game is zero-sum, A = -B. To calculate x’s payoff in a game it is: x.transpose A y. The A payoff matrix can be replaced by a function approximator -> x’s advantage function $Adv(s,a^x,a^y)$. $Adv(s,a^x,a^y)$ * x dot y is then the proper form of a two-players game payoff for x.
>       In Game Theory, we know an agent can play against itself via the optimization of “self-play”. So in each iteration, what we really have is $Adv(s,a^x,a^y)$ * x dot x. Let x be represented as a policy neural network parameterized by theta, Eq 2 can be computed as the derivative of product rule.
> 	This derivation shows the policy gradient changes in a two-player game. The discovery here is that the gradient by default is in the form of Mutual Information maximization.
>
> Q3: It’s not clear to me how mutual information helps in multi-agent specialization.
>
> 	A3: This is a great question from the Reviewer. We have seen from the above explanation that by default mutual information is maximized when two players play against each other. This led to agents behaving more and more mutually similar by default. But what does this do in a heterogeneous population joint policy training?
>       Our experiment shows Mutual Information Maximization causes “conformity behavior” in a heterogeneous population. This is harmful to agents with character attributes different from the population norm. (If we have 10 different fishes and 1 monkey in a population, the monkey will be optimized towards how to swim due to MI maximization, conformity)
> 	Besides discovering this harmful effect, we also propose M&M to minimize Mutual Information. MI minimization is to learn individualized strategy - agent specialization. This allows the monkey to form its own specialized strategy based on its own character attributes.
>       In our heterogeneous population study, we see that this benefit the performance of the population with a significant boost on the lower end of the agents (Experiment 4.5). We also see more diversified behaviors being developed (Experiment 4.2) by switching away from the default MI maximization.
>
> Q4. This work would benefit from testing on more tasks.
>
>       A4: We, too, believe more research on the learning behaviors of heterogeneous populations is needed. This would have a profound impact on the understanding of group behaviors. At this moment, we are shy of project budget to start another population study. But we are hopeful that more future research in this direction can bring this branch of research into the mainstream.
>
> We thank the Reviewer for asking the hard technical questions, it is our fault for omitting the full explanation as the paper content is approaching the page limit. We are revising the paper per the Reviewer’s feedback and suggestion such that the paper clarity is up to Reviewer’s standard. Thank you.

---

> ### Author Response · Authors · 2022-11-11
> **Second Thread: Paper Revision #1**
>
> We have prepared a point-by-point response to the questions for each Reviewer in the First Thread. (Below this thread)
>
> For our first paper revision, we have uploaded a red-line / boxed highlight revision of our paper.
>
>       The red-lines address the specific issues pointed out by the individual reviewers.
>
>       The boxed highlighter highlights portions of paper that are added to better clarify, define the undefined notations, terms, definitions of social graph and more.
>
>
> Here is what we have done on the paper revision:
>
>       1. We added introductory paragraphs to problem formulation, precise notations and functions definitions to improve - writing clarity
>
>       2. A couple of paragraphs (i.e. section 2.3, 3) have been rewritten with clean descriptions of the proposed methodology to improve - writing clarity.
>
>       3. Generalists, Specialists, social graph definitions and descriptions have been added to the paper explicitly.
>
>       4. We moved our pseudocode into the appendix for extra space in the main text.
>
>       5. We added additional details on Experiment 4.5’s analytic discussion.
>
>       6. We have extended Conclusion to summarize the paper.
>
> We are thankful for the invaluable feedback from the reviewers.
> The paper clarity has notably improved because of our discussion.
>
> For the following week, we appreciate your time and would like to seek additional feedback. We will continue to improve the paper quality until the paper is satisfactory to you.
> Thank you!

---

> > ### Comment · Reviewer_BFMo · 2022-11-16
> > **Response**
> >
> > Thank authors for the response. The reviewer would appreciate it if the authors could provide point-by-point response to the questions listed in Weakness.

---

> > > ### Author Response · Authors · 2022-11-16
> > > **Response Discussion**
> > >
> > > We thank the reviewer for the valuable feedback.
> > >
> > > We have prepared a point-by-point response to the questions listed in Weakness in the first response thread. (Below this thread)
> > >
> > > For the first PSRO issue, and typos have been revised in the red lined revision.
> > >
> > > We are excited and look forward to additional discussion.

---

### Official Review · Reviewer_3HQh · 2022-11-01

**Confidence:** 3
**Correctness:** 3
**Technical Novelty And Significance:** 3
**Empirical Novelty And Significance:** 3
**Recommendation:** 5

**Clarity, Quality, Novelty And Reproducibility:**

### Clarity and Quality

As mentioned in the previous section, the paper suffers heavily from a low clarity. A thorough rewriting might be necessary in order to make the paper conference material. The considered problem and proposed approach seem interesting, and I think that the authors can do a much better job in presenting their work.

### Novelty

The proposed approach seems to be novel, to the best of my knowledge.

### Reproducibility

There is no provided code in the present manuscript.

**Strength And Weaknesses:**

### Strengths

1. The paper considers an interesting problem which is relevant to the MARL community.
2. The key observation that the authors make provides an interesting insight into the intuition behind joint policy gradient, albeit it being a straightforward derivation.
3. The proposed algorithm seems to address the specialization issue of joint policy gradient.
4. Experimental evaluation of the method is present, and seems to prove the effectiveness of the algorithm with respect to vanilla joint policy gradient.

### Weaknesses

The most major weakness of the paper is that it is poorly written:
 1.  There are typos everywhere, from abstract to conclusion.

 2.  Definitions of mathematical objects are practically non-existent:
     - x and y, or $\Pi$, in the beginning of Section 2.3;
     - $\psi^k_i$ and $\epsilon$-Nash in Section 3.2.
     - $S: O \times O$ is said to be a function, but I do not understand what the range of the function is.

I understand that most things are left implicit, but this practice makes the paper prone to errors, since the derivations are not based on well-defined objects, and makes the paper very hard to read.

 3. The paper could benefit from additional remarks or further elaborations:
    - There should be an explicit formulation of the graph solver from (Chen et al. 2022) and its relevance to the problem considered, not just a reference.
    - What is the meaning of $Self-play$ function in the pseudocode?
    - What is the meaning of $D_{kl}(gradient_{step}(\pi_{\psi^k_\tau}))$?
    - What is the meaning of $Eval(\pi_{\psi^k_\tau}, \{ \Pi_g\}^N_{g=I})$, and why is the first index $I$ in these definitions?

**Summary Of The Paper:**

This paper considers a multi-agent reinforcement learning setup and takes an information-theoretic approach to the joint policy gradient method. The first observation is that the joint policy gradient maximizes the mutual information (MI) between the agents -- an observation that follows by definition of MI. The authors extract an important insight from the observation, namely, that the joint policy gradient is in fact maximizing the stability of social behavior, and may compromise the individual performances of the agents. In order to address this challenge, they propose Minimax MI algorithm, which specializes for individual goals by maximizing individual performance conditioned on the socially stable behaviors. The authors provide experimental evaluations of the proposed method, showcasing its ability to specialize.

**Summary Of The Review:**

In the current state, I would not recommend the paper for publication, due to the concerns I have raised in the Strengths and Weaknesses section. I believe that the observations made by the authors and their proposed algorithm are interesting and the topic is relevant to the community, but the paper needs a lot of improvement (see Strengths and Weaknesses section). With that being said, I am willing to reconsider my score, conditioned on the authors addressing the raised concerns.

---

> ### Author Response · Authors · 2022-11-07
> **First Thread: Paper Discussion With Reviewer On Paper Clarity, Writing**
>
> Q1: There are typos everywhere, from abstract to conclusion.
>
>      A1: We have received the feedback from the Reviewers and definitely can improve on the writing and bring a better reading experience for the readers. Both the clarity and defining the notation of the writing will be our revision focus.
>
> Q2: Definitions of mathematical objects are practically non-existent:
> $S: O \times O$ is said to be a function, but I do not understand what the range of the function is.
>
>      A2: The function’s range is to take the observation of two players and concatenate them together. e.g. Fully observable for both players.
>
> Q3: I understand that most things are left implicit, but this practice makes the paper prone to errors, since the derivations are not based on well-defined objects, and makes the paper very hard to read.
>
>      A3: We have received the feedback. Sorry about the poor writing. We will revise the paper so that notation and symbols in the derivation are explicitly defined.
>
> Q4.i.: There should be an explicit formulation of the graph solver from (Chen et al. 2022) and its relevance to the problem considered, not just a reference.
> 	A4.i.: Noted, the explicit formation will likely be placed in appendix due to page constraint.
> Q4.ii.: What is the meaning of $Self-play$ function in the pseudocode?
> 	A4.ii.: $Self-play$ function runs the game simulation on the input agents and their interaction graph with the other mixed strategy policies.
>
> Q4.iii. What is the meaning of $D_{kl}(gradient_{step}(\pi_{\psi^k_\tau}))$?
>
>      A4.iii.: This step is to constrain the gradient via PPO clipping. We will fix the notation and description will be provided.
>
> Q4.iv.: What is the meaning of $Eval(\pi_{\psi^k_\tau}, { \Pi_g}^N_{g=I})$, and why is the first index $I$ in these definitions?
>
>      A4.iv.a.: The Eval function computes the win / loss between \pi_{\psi^k_\tau} and  { \Pi_g}^N_{g=I}). The win and loss values are used to update the interaction probability between agents for the next game.
>      A4.iv.b.: We denote the population from {i, ii.. N} in the pseudo-code, and we use i, ii.. as agent IDs to index the population.
>
>  We understand the Reviewer’s concern on the notation, function and the graph being implicitly defined makes paper prone to errors. We agree with the Reviewer’s paper review standard and we are thankful for the feedback. The paper revision will improve on the above risen concerns.

---

> > ### Comment · Reviewer_3HQh · 2022-11-25
> > **Notation and clarity might still need improvement**
> >
> > Several things have improved about the clarity and notation of the paper. However, many more things remain.
> >
> > In Section 2.3, the value function is still nowhere to be found. Furthermore, in Section 3.2, the authors introduce a maximization problem, but it is not clear what the maximization variable is, although one might assume that it is $\psi^k$. The authors make use of the value function, Q-function, and advantage function in Section 2.3, all of which presume an underlying sequential scheme and such a framework is only defined later in Section 3.1. This makes the readability of the paper an unnecessarily confusing task.
> >
> > On the other hand, the integration in the Section A.6 of the Appendix is not rigorous, and the end result is a tautology: $\log (\Pi_\theta^x(a^x|s)\Pi_\theta^y(a^y|s)) = \log (\Pi^x_\theta (a^x|s)\Pi^y_\theta(a^y|s))$. I do not understand why the integration is needed if you conclude that this is equal to $\log ( \Pi^{(x,y)}_\theta (a^x,a^y|s))$ by independence. Also, citing a Stack Exchange result (which the authors do in this section) does not seem appropriate for a scientific paper.
> >
> > Based on the aforementioned remaining concerns, I keep my score as it is, and encourage the authors to revise the paper further.

---

> > > ### Author Response · Authors · 2022-11-29
> > > **Appreciation to Reviewer**
> > >
> > > We thank the Reviewer for your time, suggestion and feedback.
> > >
> > > We will keep the feedback in mind and improve on the paper.
> > >
> > > Thank you.

---

> ### Author Response · Authors · 2022-11-11
> **Second Thread: Paper Revision #1**
>
> We have prepared a point-by-point response to the questions for each Reviewer in the First Thread. (Below this thread)
>
> For our first paper revision, we have uploaded a red-line / boxed highlight revision of our paper.
>
>       The red-lines address the specific issues pointed out by the individual reviewers.
>
>       The boxed highlighter highlights portions of paper that are added to better clarify, define the undefined notations, terms, definitions of social graph and more.
>
>
> Here is what we have done on the paper revision:
>
>       1. We added introductory paragraphs to problem formulation, precise notations and functions definitions to improve - writing clarity
>
>       2. A couple of paragraphs (i.e. section 2.3, 3) have been rewritten with clean descriptions of the proposed methodology to improve - writing clarity.
>
>       3. Generalists, Specialists, social graph definitions and descriptions have been added to the paper explicitly.
>
>       4. We moved our pseudocode into the appendix for extra space in the main text.
>
>       5. We added additional details on Experiment 4.5’s analytic discussion.
>
>       6. We have extended Conclusion to summarize the paper.
>
> We are thankful for the invaluable feedback from the reviewers.
> The paper clarity has notably improved because of our discussion.
>
> For the following week, we appreciate your time and would like to seek additional feedback. We will continue to improve the paper quality until the paper is satisfactory to you.
> Thank you!

---

### Official Review · Reviewer_LEQx · 2022-11-02

**Confidence:** 2
**Correctness:** 4
**Technical Novelty And Significance:** 3
**Empirical Novelty And Significance:** 3
**Recommendation:** 6

**Clarity, Quality, Novelty And Reproducibility:**

Clarity: Motivation and methodology part of this paper is very clear. I appreciate the authors efforts in well drafting the related work section. I wish there was a bit more discussion for qualitative analysis in experiments section.

Quality/Novelty: minimax formulation of mutual information for agents specialization sound to be novel in multi-agent learning settings. I wish there was more space in the main paper to expand on experimental details for some of the sections and qualitative analysis, but the paper is quite content-full as is.


**Strength And Weaknesses:**

Authors have well drafted the related work/background section, clearly explaining past work in competitive & cooperative behavioral learning. Solid motivation pointing out the trade offs with MI maximization/minimization alone. Gains in empirical evaluation are significant - performance gain from agent specification is significant(few agents beating sota by >20 points interms of winrate%). Also, 30 % reduction in population performance deviation, as authors pointed out, shows that equity of a population can be improved when individual agents specialize.

Some sub-sections under experiment aren’t clear/seek more details. In section 4.5, Winrate % is not consistent across all agents, specifically seems to be more beneficial for weaker agents. For example, agent T, agent D see no/little performance gain with M&M specialization. Also, training efficiency and competitiveness is not clear and require more details - for ex, why are benchmarks limited/specific to 2 agents (K and M)? Lastly, what would the impact of number of unique agents on joint policy optimization benefits (in these experiments, 16 unique agents of Naruto Mobile characters were chosen).

**Summary Of The Paper:**

Given Joint policy optimization encourages knowledge sharing and social behaviors, this paper connects the dots between policy gradient formulation of joint policy optimization and MI maximization. Authors have proposed a minimax formulation of MI (M&M) that enables agents specialization with stable regularization. It’s pointed as a good sweet spot to balance the tradeoffs with MI maximization/minimization alone - M&M optimizes agent specialization while regularizing the learning instability with conditional MI minimization. Authors have included a theoretical and experimental analysis on the social behavioral change of agent population and demonstrate that M&M benefits in population social diversity and specialists competitive play.

**Summary Of The Review:**

The motivation of this paper is technically sound and empirical evaluations shown significant performance gains. This paper features minimax formulation of MI (M&M) that enables agents specialization with stable regularization. experimental results have shown significant benefits in population social diversity and specialists competitive play. However, some sub-sections of the experimental section aren't clear/seek more details. It would have been further helpful if authors included qualitative analysis. For more details, please refer to the strengths and weaknesses part of the review.

---

> ### Author Response · Authors · 2022-11-07
> **First Thread: Paper Discussion With Reviewer On Why Performance Gains Are Inconsistent**
>
> Q1: Some sub-sections under experiment aren’t clear/seek more details. In section 4.5, Winrate % is not consistent across all agents, specifically seems to be more beneficial for weaker agents. For example, agent T, agent D see no/little performance gain with M&M specialization.
>
>       A1: Right, we definitely can do better to improve the experiment with clarity and details. From the joint policy optimization to M&M, we believe the inconsistency is part of the problem that M&M is addressing.
>
>       Inconsistency: Joint policy optimization has gradients that implicitly maximize mutual information from Eq 1~3. This means the optimization focuses on commonality between all pairs of agents. This process is “harmful” to agents that have character attributes that fall outside of the population norm. Agent K and Agent A are not weak per say, their attributes are unique and do not benefit from mutual behavior optimization with the average population. M&M lifts the mutual information maximization constraint. We see that the unique agents benefit from M&M’s specialization much more. Agent T and D character attributes seem to be close to the population norm. In this case, it becomes difficult to further specialize as the underlying character attributes are not unique.
>
> Q2: Also, training efficiency and competitiveness is not clear and require more details - for ex, why are benchmarks limited/specific to 2 agents (K and M)?
>
> 	A2: Agent K is one of the agent with “weaker” performance, and Agent M is an agent that already performs relatively well with joint policy optimization. We selected the two agents to show a clean comparison of training efficiency between a “weaker” agent and a “stronger” agent.
>
> Q3: Lastly, what would the impact of number of unique agents on joint policy optimization benefits (in these experiments, 16 unique agents of Naruto Mobile characters were chosen).
>
>       A3: This is a good question. We believe it would heavily depend on the character attribute distributions of the added / subtracted agents. Since the character attribute is internal to the game engine, we have not found a way to quantify the attributes data so they can be collected and visualized. In NeuPL Chen et al. 2022’s paper they have done a population size study on this question (page 7 of their paper). Their experiment is on whether a new strategy is created from increasing the maximum neural population size. We believe the study brings insight to the neural net capacity for learning, but also faces the same question we have: What is the distribution of the added character attributes? Without knowing the distribution it is difficult to study whether the bottleneck is the neural net or the lack of diversity of the population.
>
> We are happy that the Reviewer and the community find analyzing population behavior in this manner interesting. We thank the Reviewer for the helpful feedback. Q1 and Q2 are especially important questions that help us pinpoint the specific places to improve the clarity of the paper. Thank you!

---

> ### Author Response · Authors · 2022-11-11
> **Second Thread: Paper Revision #1**
>
> We have prepared a point-by-point response to the questions for each Reviewer in the first response thread. (Below this thread)
>
> For our first paper revision, we have uploaded a red-line / boxed highlight revision of our paper.
>
>       The red-lines address the specific issues pointed out by the individual reviewers.
>
>       The boxed highlighter highlights portions of paper that are added to better clarify, define the undefined notations, terms, definitions of social graph and more.
>
>
> Here is what we have done on the paper revision:
>
>       1. We added introductory paragraphs to problem formulation, precise notations and functions definitions to improve - writing clarity
>
>       2. A couple of paragraphs (i.e. section 2.3, 3) have been rewritten with clean descriptions of the proposed methodology to improve - writing clarity.
>
>       3. Generalists, Specialists, social graph definitions and descriptions have been added to the paper explicitly.
>
>       4. We moved our pseudocode into the appendix for extra space in the main text.
>
>       5. We added additional details on Experiment 4.5’s analytic discussion.
>
>       6. We have extended Conclusion to summarize the paper.
>
> We are thankful for the invaluable feedback from the reviewers.
> The paper clarity has notably improved because of our discussion.
>
> For the following week, we appreciate your time and would like to seek additional feedback. We will continue to improve the paper quality until the paper is satisfactory to you.
> Thank you!

---

### Official Review · Reviewer_fF3t · 2022-11-02

**Confidence:** 3
**Correctness:** 2
**Technical Novelty And Significance:** 3
**Empirical Novelty And Significance:** 4
**Recommendation:** 6

**Clarity, Quality, Novelty And Reproducibility:**

The paper is not easy to follow (even for me who is fairly well integrated in this field). The paper has several typos which I have tried to overlook in this review, but urge the reviewers to check carefully.  I did not find any mention about the code. So it is unclear whether the solution is reproducible.

**Strength And Weaknesses:**

Pros:
1. This study is a valiant attempt to demystify multi-agent learning in current RL models
2. The proposed solution is intuitively easy to understand.

Cons:
1. The paper is really hard to follow and there are several typos.
2. It is hard to know if the experiments truly generalize. Besides the theoretical argument, the experiments were performed on a single game: Naruto Mobile. While the authors report considerable improvement there is considerable work to be done.
3. There is an obvious link between stability (specilization) and plasticity (generalization to new tests). The authors have not discussed this theme. It is not always desirable for agents in a multi-agent game to specialize and I wonder how the authors think about this in context of their work. I worry that M&M may NOT in fact the most desirable in most case of AI based training methods.
4. The conclusion and discussion sections were notably sparse on details.


**Summary Of The Paper:**

The paper deal with the problem of mutii-agent specialization: which is the issue related to the developing of a single general learning algorithms for a relatively diverse population. Typically each agent in a multi-agent paradigm is trained using separate behavioral policies. But this has now led way to jointly optimized learning solutions,

The authors here report that the joint policy is related to a simple information theoretic measure: mutual information (MI) among agents. This solution however leads to a bottleneck, which is that individual agents do not specialize. To circumvent this issue, the authors derive a minimax MI solution (M&M) and show that the agent trained using this method not are better specialisists but do better on competitive play metrics.


**Summary Of The Review:**

The paper makes an interesting observation regarding the training of multi agent models. The proposed solution however seems rather extreme and the context under which this would be relevant in the training of other multi-agent models remains to be seen. The current results on a single benchmark are well taken.

---

> ### Author Response · Authors · 2022-11-07
> **First Thread: Paper Discussion With Reviewer On Plasticity And Specialization In A Population**
>
> Q1: There is an obvious link between stability (specialization) and plasticity (generalization to new tests). The authors have not discussed this theme. It is not always desirable for agents in a multi-agent game to specialize and I wonder how the authors think about this in context of their work. I worry that M&M may NOT in fact the most desirable in most case of AI based training methods.
>
>       A1: This is an important and not easy to answer question on specialization and plasticity. We can see the argument going in both directions. In our opinion, we subscribe to the economic idea of Adam Smith of the Division of Labor.  In a heterogeneous population, like our research study, each agent is innately better at performing some skills, policy than the others. When each agent specializes on what they are good at relative to the general population (Generalist), the performance of the whole population can be further increased as shown in Experiment 4.5. This is the background philosophy of our paper. Of course, we also see the Reviewer's point of view that continuing to stay plastic, and adapt to any problem on hand is also an essential quality in learning too. This is a similar exploration (plastic) and exploitation (specialization) dilemma on a population scale. We wonder if the Reviewer has a different view on the dilemma.
>
> Q2: The paper is really hard to follow and there are several typos.
>
>       A2: We are working on improving the clarity and the writing of the paper, and hope that we can bring the readers a better reading experience.
>
> Q3: It is hard to know if the experiments truly generalize. Besides the theoretical argument, the experiments were performed on a single game: Naruto Mobile. While the authors report considerable improvement there is considerable work to be done.
>
>       A3: We have seen another Reviewer who also pointed out that evaluating additional games may be helpful. At the given moment, we may not have the project budget for another population study, but we are hopeful that when sample efficiency improves, future research in population behaviors will become more and more mainstream.
>
> Q4: The conclusion and discussion sections were notably sparse on details.
>
>       A4: Noted, we are adding more details such as our discussion into the conclusion and discussion for the paper revision.
>
> We thank the Reviewer for the paper feedback and also would love to continue with the plasticity and specialization discussion. The revision of the paper will focus on improving the clarity of the paper and bring a better reading experience.

---

> ### Author Response · Authors · 2022-11-11
> **Second Thread: Paper Revision #1**
>
> We have prepared a point-by-point response to the questions for each Reviewer in the first response thread. (Below this thread)
>
> For our first paper revision, we have uploaded a red-line / boxed highlight revision of our paper.
>
>       The red-lines address the specific issues pointed out by the individual reviewers.
>
>       The boxed highlighter highlights portions of paper that are added to better clarify, define the undefined notations, terms, definitions of social graph and more.
>
>
> Here is what we have done on the paper revision:
>
>       1. We added introductory paragraphs to problem formulation, precise notations and functions definitions to improve - writing clarity
>
>       2. A couple of paragraphs (i.e. section 2.3, 3) have been rewritten with clean descriptions of the proposed methodology to improve - writing clarity.
>
>       3. Generalists, Specialists, social graph definitions and descriptions have been added to the paper explicitly.
>
>       4. We moved our pseudocode into the appendix for extra space in the main text.
>
>       5. We added additional details on Experiment 4.5’s analytic discussion.
>
>       6. We have extended Conclusion to summarize the paper.
>
> We are thankful for the invaluable feedback from the reviewers.
> The paper clarity has notably improved because of our discussion.
>
> For the following week, we appreciate your time and would like to seek additional feedback. We will continue to improve the paper quality until the paper is satisfactory to you.
> Thank you!

---

### Official Review · Reviewer_khP1 · 2022-11-03

**Confidence:** 4
**Correctness:** 2
**Technical Novelty And Significance:** 2
**Empirical Novelty And Significance:** 2
**Recommendation:** 5

**Clarity, Quality, Novelty And Reproducibility:**

* There are some minor issues (see weaknesses), in terms of writing that needs to be further improved.
* The main idea is only marginally significant and not well-supported by experiments and also lack of a comparison with state-of-the-art models.

**Details Of Ethics Concerns:**

There is no ethical issue.

**Strength And Weaknesses:**

The most important merit of this paper is the contribution toward an equilibrium between generalization and expertise in multi-agent systems without having any assumptions about cooperation or competitive setup. Once the initialization happens, the proposed algorithm will basically follow the min-max strategy to reach an equilibrium satisfying both generalization and expertise.

**Strengths**
* No assumption on the cooperation/competition.
* Practical on policy optimization that provides better performance compared to value-based methods.
* Intuitive representation of results using radial charts.

**Weaknesses**
* The contributions are not highlighted enough in the paper.
* Although it cannot be a direct weakness, we should try to avoid using KL in policy optimization due to complexity.
* Background and Results are limited to two player games.
* The results are not compared to any state-of-the-art method.
* The algorithm is not discussed in details.


**Summary Of The Paper:**

As implied by the author, multi-agent systems can be studied in the sense of social behavior. Where each agent needs to be an expert in a particular task, they should share a common understanding of the whole space with other agents. To this end, a wide variety of methods are proposed to fulfill both of these criteria. One can think of that common understanding/knowledge as the ability to generalize from bottom to top. That means when an agent is an expert in one task, it can easily be extended to contribute to the solution of a larger task. Since an intelligent agent can be showcased by using a single policy function, many of the existing methods in this matter focus on relations between different agents(a.k.a policies). This paper generally centers around proposing an optimization framework in multi-agent systems by using Reinforcement Learning notions and classic game theory. Specifically, by using the Mutual Information concept, which conveys the level of shared understanding of the same random variable, a gap will rise between generalization and expertise. Having grounds of knowledge about almost everything will, obviously, be counterproductive in the case of getting narrow in a specific task. Therefore, the author tries to employ classic game theory to balance this dilemma.

**Summary Of The Review:**

* As stated before, the contribution is not perfectly annotated in the paper. The author pretty much employs the related techniques in the proposed algorithm and does not clearly claim the novelty.
* The proposed method closes an essential gap in multi-agent systems, but still the generalist seems to take so much time to converge
* The overall quality of paper sounds acceptable, but slight improvements on the presentation, specifically, on the contribution will make it more understandable

---

> ### Author Response · Authors · 2022-11-11
> **Paper Revision #1**
>
> We have included many suggestions from your constructive feedback into our revised paper.
> For our first paper revision, we have uploaded a red-line / boxed highlight revision of our paper.
>
>       The red-lines address the specific issues pointed out by the individual reviewers.
>
>       The boxed highlighter highlights portions of paper that are added to better clarify, define the undefined notations, terms, definitions of social graph and more.
>
>
> Here is what we have done on the paper revision:
>
>       1. We added introductory paragraphs to problem formulation, precise notations and functions definitions to improve - writing clarity
>
>       2. A couple of paragraphs (i.e. section 2.3, 3) have been rewritten with clean descriptions of the proposed methodology to improve - writing clarity.
>
>       3. Generalists, Specialists, social graph definitions and descriptions have been added to the paper explicitly.
>
>       4. We moved our pseudocode into the appendix for extra space in the main text.
>
>       5. We added additional details on Experiment 4.5’s analytic discussion.
>
>       6. We have extended Conclusion to summarize the paper.
>
> We are thankful for the invaluable feedback from the reviewers.
> The paper clarity has notably improved because of our discussion.
>
> For the following week, we appreciate your time and would like to seek additional feedback. We will continue to improve the paper quality until the paper is satisfactory to you.
> Thank you!

---

> > ### Comment · Reviewer_khP1 · 2022-11-17
> > **Good improvements**
> >
> > **Improvements**
> > * Providing more specific information on the contribution
> > * Elaboration on proposed method
> > * Explanation on the notations and algorithm
> > * The conclusion section reflects the method with respect to the results
> > * The overall clarity of the paper is much better due to rearranging some sections and providing the above mentioned information
> >
> > **Comments and questions**
> > * In Fig. 4, agent T is not reflecting the proposed method’s performance. It is better to use standard vertical bar charts with 2 bars per agent to reflect the details
> > * It is better to provide an outline of your experiments in section 4 (e.g., our experiments include behavioral analysis, advantage analysis and …) and discuss the necessity of each experiment. You can then elaborate on details of each experiment in the respective section.
> > * I find two experiments which showcase the performance, but specifically Fig 5, is not informative with the vague caption and discussion provided.
> > * Did you analyze the bottleneck of your method which takes much time compared to other steps? Is it the EVAL step? NPL step?
> >
> > In general, the paper is now much improved in terms of clarity and quality. Despite the narrow contribution presented in the paper, the study itself is valuable as a new perspective in joint policy optimization.

---

> > > ### Author Response · Authors · 2022-11-17
> > > **Paper Discussion and Minor Paper Revision**
> > >
> > > We appreciate the Reviewer for the 2nd round of feedback. The current version of the paper has been revised to improve on the above mentioned points
> > >
> > > Discussion and Comments:
> > >
> > >       1. Noted. Using 2 bars per agent is a great idea! This would definitely help to clarify the case for Agent T. We have revised Fig. 4 accordingly.
> > >
> > >       2. We have also taken the Reviewer's advice on adding an short outline of our experiments as part of the intro in section 4.
> > >
> > >       3. + 4. Are related to training sample efficiency.
> > >       We revised the vague caption and discussion in section 4.6 to provide more details on the issue. Short description: Since M&M optimization no longer optimize with MI maximization as an objective, each experience sample is unique to each agent and would create separate learning distribution for each agent. To collect each separated learning experiences, additional computation is required.
> > >
> > > We thank the Reviewer.
> > > Your feedback is invaluable and have helped us to improve the paper in more than one way. Thank you!

---

> > > > ### Comment · Reviewer_khP1 · 2022-11-19
> > > > **Concerns are clear**
> > > >
> > > > Thank you very much for your time addressing my questions and concerns! I believe that the paper has modest contribution to the RL field, so I raised the score to 6.

---

> > > > > ### Author Response · Authors · 2022-11-19
> > > > > **Appreciation to Reviewer**
> > > > >
> > > > > The pleasure is all ours. Thank you very much for giving our paper a second chance.
> > > > >
> > > > > We greatly appreciate it.

---

### Official Review · Reviewer_NL8L · 2022-11-03

**Confidence:** 2
**Correctness:** 3
**Technical Novelty And Significance:** 3
**Empirical Novelty And Significance:** 3
**Recommendation:** 6

**Clarity, Quality, Novelty And Reproducibility:**

Clarity and quality:
The motivation and related work sections are well-written, but the writing of the proposed method is unclear. There are several typos in the paper.

Novelty:
In my perspective, the approach has a certain novelty in it.

Reproducibility:
The source code is not provided and the way the method is introduced makes it challenging to reproduce the results solely based on the paper.

**Strength And Weaknesses:**

Strengths: The key idea is straightforward and easy to understand. The paper is backed by a succinct theoretical analysis when it talks about the underlying relationship between joint policy gradient and MI maximization.

Weaknesses: Even though the method is well-motivated, the proposed method is not easy to follow, with terms and symbols left undefined. Moreover, the experiment section needs more work. I would like to see a more thorough evaluation and clearer explanation of the M&M results compared to NeuPL, such as the root cause of Agent A’s significant improvement. In addition, it would be good to show how M&M works in games with more cooperation among agents like MuJoCo Football mentioned in the NeuPL paper.


**Summary Of The Paper:**

This paper offers a new perspective to solve the behavior homogeneity issue in current joint policy optimization approaches. The paper points out the connections between policy gradient in multi-agent joint policy optimization and mutual information (MI) maximization followed by the drawbacks of MI maximization. The paper proposes a minimax formulation of MI, M&M, to improve agents’ diversity and specialization while maintaining the population stability.

**Summary Of The Review:**

This paper purposes a minimax MI approach to mitigate the behavior homogeneity issue among agents in joint policy optimization. The idea proposed in the paper could be inspiring to the community, but I would expect a better presentation of the proposed method and a more comprehensive evaluation and discussion.

---

> ### Author Response · Authors · 2022-11-07
> **First Thread: Paper Discussion With Reviewer On Evaluation Improvement**
>
>
> Q1: Even though the method is well-motivated, the proposed method is not easy to follow, with terms and symbols left undefined.
>
>       A1: Right. The notations, symbols and terms are left undefined. This is an oversight on our part. A paper revision is on the way to improve this aspect and the clarity on the proposed method.
>
> Q2: Moreover, the experiment section needs more work. I would like to see a more thorough evaluation and clearer explanation of the M&M results compared to NeuPL, such as the root cause of Agent A’s significant improvement.
>
>       A2: The root cause of Agent A and Agent K’s significant improvement is definitely an interesting question. The difference from joint policy MI maximization to Conditioned MI minimization, is that Agent A and Agent K are free to behave differently from the population norm. In other words, when MI is maximizing, all agents’ behaviors are mutually similar. This is a heavy constraint on agents that are innately different from the average population. When the constraint is lifted with M&M, agents are free to specialize based on their innate attributes.
>
> Q3:  In addition, it would be good to show how M&M works in games with more cooperation among agents like MuJoCo Football mentioned in the NeuPL paper.
>
>       A3: We would love to see this in action. Cooperation element definitely adds another dynamic to setting where the population is heterogeneous. At the given moment, we may not have sufficient project budget for this population study, but we are hopeful that this would be a promising future research direction.
>
>
> We thank the many insightful and interesting questions posed by the Reviewer. Reviewer has provided us with helpful feedback to strengthen the clarity and evaluation of the paper. A paper revision is on the way and will improve on these aspects. Thank you!

---

> ### Author Response · Authors · 2022-11-11
> **Second Thread: Paper Revision #1**
>
> We have prepared a point-by-point response to the questions for each Reviewer in the first response thread. (Below this thread)
>
> For our first paper revision, we have uploaded a red-line / boxed highlight revision of our paper.
>
>       The red-lines address the specific issues pointed out by the individual reviewers.
>
>       The boxed highlighter highlights portions of paper that are added to better clarify, define the undefined notations, terms, definitions of social graph and more.
>
>
> Here is what we have done on the paper revision:
>
>       1. We added introductory paragraphs to problem formulation, precise notations and functions definitions to improve - writing clarity
>
>       2. A couple of paragraphs (i.e. section 2.3, 3) have been rewritten with clean descriptions of the proposed methodology to improve - writing clarity.
>
>       3. Generalists, Specialists, social graph definitions and descriptions have been added to the paper explicitly.
>
>       4. We moved our pseudocode into the appendix for extra space in the main text.
>
>       5. We added additional details on Experiment 4.5’s analytic discussion.
>
>       6. We have extended Conclusion to summarize the paper.
>
> We are thankful for the invaluable feedback from the reviewers.
> The paper clarity has notably improved because of our discussion.
>
> For the following week, we appreciate your time and would like to seek additional feedback. We will continue to improve the paper quality until the paper is satisfactory to you.
> Thank you!

---

> > ### Comment · Reviewer_NL8L · 2022-11-24
> > **Improvement on clarity**
> >
> > The clarity of the paper is improved with the revisions. There are more elaborations on the proposed method and its contributions. I would like to keep my score of marginally above the acceptance threshold.

---

> > > ### Author Response · Authors · 2022-11-25
> > > **Appreciation to Reviewer**
> > >
> > > We thank the Reviewer for your feedback and support. Thank you.

---

### Official Review · Reviewer_ryJg · 2022-11-04

**Confidence:** 3
**Correctness:** 3
**Technical Novelty And Significance:** 3
**Empirical Novelty And Significance:** 2
**Recommendation:** 3

**Clarity, Quality, Novelty And Reproducibility:**

- The technical section of the paper is a bit difficult to read particularly because the terms used are not well defined and some notations are ambiguous. Please see the list below :

    - Is $\{\Pi^g_{\theta^\*}\}_g$ a solution policy set obtained from $\epsilon$\-Nash joint policy optimization or is it the optimization problem itself, first line of section 3.2?

    - The non-Markov two-player game setting is not well specified - how is agents utility $U$ different from reward $R$ ? What does weighted edges represent and what constitutes an adversarial interactions discussed just above section 3.2?

    - $[0:1]$ is a python notation; better to use $\gamma \in [0,1)$.

    - Multiplying a policy $\Pi^g_{\theta^\*}$ which is a function of $\mathcal O \times \mathcal A$, to an expectation term as in equation 7 is ambiguous.


    - How does equation $(3)$/(8) reduces to the form in equation $(4)$/(9) respectively? how did the $Q$ and $V$ function vanish? The statement after eqn (3) “With T steps of optimization, the integration of gradients results to a long-term behavioral change of mutual information maximization” ([pdf](zotero://open-pdf/library/items/ER8E6IYM?page=3)) is also not very clear.

    - This statement is not clear “The formulation can be extended to any two-player exchange of (X,Y) of N distinct agents {i,ii,..N}. MI is expanded as Interaction Information”.

**Strength And Weaknesses:**

- The paper is well motivated and seeks to learn agents with diversified behavior pattern in a multi-agent policy optimization framework.

- The experimental section demonstrated the advantage of their algorithm over prior works.

- However, the technical sections 2.3 and 3 lacks clarity and has some issues(see comments in next section for more details).

**Summary Of The Paper:**

The paper connects policy gradient method in joint policy optimization of multi-agent RL to MI maximization framework and shows that joint policy optimization essentially leads to MI maximization. Thus joint policy optimization produces skills that are transferable across agents but it costs in diversity and performance of agents. To address this issue, the authors propose a novel minimax algorithm called M&M that is able to learn agents that specialize at different skill via conditioning on the stable generalist policies obtained through MI regularization. Experimental results demonstrate the similarity between the agents learned through joint policy optimization and how specialist agents learned by their algorithm shift away from Generalist’s behavior to specialize in their own specific behaviors. They also presented a qualitative comparison of their algorithm with prior works like NeuPL.

**Summary Of The Review:**

The paper is well motivated and the proposed M&M algorithm appears quite novel. However, technical section 2.3 and section 3 lacks clarity and should be improved further.

---

> ### Author Response · Authors · 2022-11-06
> **First Thread: Paper Discussion With Reviewer**
>
> Clarity, Quality, Novelty And Reproducibility:
>
> Q1: Is ${\Pi^g_{\theta^*}}_g$ a solution policy set obtained from $\epsilon$-Nash joint policy optimization or is it the optimization problem itself, first line of section 3.2?
>
> 	A1: ${\Pi^g_{\theta^*}}_g$ is the solution policy set obtained from $\epsilon$-Nash joint policy optimization. This is the solution set that is based on the Mutual Information Maximizing formulation from Eq 1 to Eq 3.
>
> Q2: The non-Markov two-player game setting is not well specified - how is agents utility $U$ different from reward $R$ ? What does weighted edges represent and what constitutes an adversarial interactions discussed just above section 3.2?
>
>          A2.i.: We define the utility, $U$, on the result of the games, i.e. a value of 1 for win or value of 0 for a loss. Based on $U$ of the prior matches between agents, graph solver $F$ updates the probability of agents' matchup to other agents. The reward $R$, on the other hand, is used for action policy learning within a game that may include learning signal such as time left, resources used/gained during a game.
>           A2.ii.: $U$ are directional weighted edges. The higher the weighted value of vertices a-to-b, the higher a's winrate is to b.
>           A2.iii.: Adversarial interactions is to let $F$ to optimize the matchup such that strongest adversary set is sampled for the next match. (Other implementation such as closest strength matching is also possible)
>
> Q3: [0:1] is a python notation; better to use $\gamma \in [0,1)$. & Multiplying a policy $\Pi^g_{\theta^*}$ which is a function of $\mathcal O \times \mathcal A$, to an expectation term as in equation 7 is ambiguous.
> 	A3: Right, a revision will include a correction to address the notations
>
> Q4: How does equation (3)/(8) reduces to the form in equation (4)/(9) respectively? how did the $Q$ and $V$ function vanish? The statement after eqn (3) “With T steps of optimization, the integration of gradients results to a long-term behavioral change of mutual information maximization” (pdf) is also not very clear.
>
>           A4.i.: We have taken out $Q$ and $V$ so it is easier to spot the Mutual Information equation. Reading it without $Q$ and $V$ definitely hurts the clarity. We thank the reviewer for the feedback, eq 8 and 9 will be revised as weighted Mutual Information with $Q$ and $V$.
>
> 	A4.ii.: Eq 3’s form is similar to, but still not Mutual Information maximization - the “delta” is still left in the equation. The delta vanishes through T steps of such optimization (integration from 0 to T). So the derivation is saying that Mutual Information maximization occurs between agents when agents have long-term interactions. The accumulated behavioral change over time makes them more and more mutually similar even if they are enemies (competing).
>
> Q5.:This statement is not clear “The formulation can be extended to any two-player exchange of (X,Y) of N distinct agents {i,ii,..N}. MI is expanded as Interaction Information”.
>
> 	A5: This statement needs a correction. We thank the reviewer for pointing it out.
>
>
> We are delighted that the Reviewer finds the paper interesting, and pointed out specific aspects of the paper for improvement. The feedback and advise are helpful and definitely help us make our paper better. Please stay tuned, a revision of the paper is on the way. We thank for the Reviewer’s time and helpful feedback.
>
> ----------------------------------------------------------------------------
> We have included many suggestions from your constructive feedback into our revised paper.
> For our first paper revision, we have uploaded a red-line / boxed highlight revision of our paper.
>
>       The red-lines address the specific issues pointed out by the individual reviewers.
>
>       The boxed highlighter highlights portions of paper that are added to better clarify, define the undefined notations, terms, definitions of social graph and more.
>
>
> Here is what we have done on the paper revision:
>
>       1. We added introductory paragraphs to problem formulation, precise notations and functions definitions to improve - writing clarity
>
>       2. A couple of paragraphs (i.e. section 2.3, 3) have been rewritten with clean descriptions of the proposed methodology to improve - writing clarity.
>
>       3. Generalists, Specialists, social graph definitions and descriptions have been added to the paper explicitly.
>
>       4. We moved our pseudocode into the appendix for extra space in the main text.
>
>       5. We added additional details on Experiment 4.5’s analytic discussion.
>
>       6. We have extended Conclusion to summarize the paper.
>
> We are thankful for the invaluable feedback from the reviewers.
> The paper clarity has notably improved because of our discussion.
>
> For the following week, we appreciate your time and would like to seek additional feedback. We will continue to improve the paper quality until the paper is satisfactory to you.
> Thank you!

---

> > ### Comment · Reviewer_ryJg · 2022-11-25
> > **Significant technical issues in section 2 and 3; needs more revision.**
> >
> > Thanks for updating the paper. I have some more questions regarding the setting and the claims made in technical sectiion of the paper. I would appreciate if you can provide a point by point response to my queries.
> >
> > 1. In competitive setting, why should the players maximize joint expected return $J(\theta)$ as stated in section 2.3? Usually, in competitive setting each players cares to maximize its own returns and jointly reaching something sensible like Nash equilibrium is considered ideal for rational agents. Can you clarify why does the objective of joint value optimization make sense in here?
> >
> > 2. The proof for equivalence between MI maximization and optimizing $J(\theta)$ as in eqn 2 is not very convincing in particular because in your definition of $J(\theta)$ the critic is constant over time which is not true(even in practice $Q$ is kept fixed only for a few iteration and is then updated through an critic update). However this aspect has not been taken into account in your equivalence proof.
> >
> > 3. I am also not very convinced the redefinition given in eqn 4. I doubt that for a general $w$ this notion of “weighted MI” would have the nice properties of mutual information. Is it even symmetric? Just calling it MI based on the form it takes in not justified. Can you justify it?
> >
> > 4. The statement about the convergence of join policy(after eqn 3) is also not very clear. The integration explanation in Appendix A.6 is a very trivial statement about logarithm function $\log(xy) = \log(x) + \log(y)$. I don’t see how does it say anything about the convergence of $\theta$.
> >
> > 5. Why the heuristic of pairing the agents and summing the gradients over all pair of agents(as in eqn 5) makes sense as long as the joint value objective is concerned?
> >
> > 6. The mutual information expression is the gradient of $J(\theta)$ objective : can you clarify how does this statement follow : “the maximization of $J(\theta)$ results in MI maximization in the behavior of a heterogeneous population”?
> >
> > 7. The BR response strategy of players need not even converge(for example in rock, scissor, paper game, if each each player best responds to strategy of players would never converge). So, the claim that M&M method leads to agents with specialized innate attributes(before section 3.3) is not well justified. Why does innate even mean here?
> >
> > Minor notation issue : Please rename the agent set from $\{i, ii, …, N\}$ to $\{1,2,,…,N\}$ to avoid confusing romanized 1 with variable alphabet $i$.

---

> > > ### Author Response · Authors · 2022-11-26
> > > **7 Critical Discussions With Reviewer**
> > >
> > > We thank the Reviewer for another round of great queries. Here are the point-by-point responses:
> > >
> > > Q1. "Usually, in a competitive setting each player cares to maximize its own returns... why does the objective of joint value optimization make sense in here?"
> > >
> > >     A1: In the context of multiagent optimization, the opponents may not be always accessible. For example, Alphago had to play against itself, rather than Mr. Lee Sedol during its training. The hope is that the policy behaviors learned during self-play would transfer during inference.  Joint policy optimization is to scale the concept of self-play to a population level. A population competes against itself such that the learned joint policy may have competitive behaviors during inference. (In our case, we may deploy the agents to play against real world user.) So jointly optimize all policies together is to address the weakness in one's self before playing against some unknown distribution of opponent in future inference.
> > >
> > > Q2. "...definition of $J(\theta)$ the critic is constant over time which is not true..."
> > >
> > >     A2: The Reviewer's observation is correct, the change of critic is not accounted in our derivation. Our derivation focuses on the policy gradient part of the joint policy optimization. After each iteration of critic optimization, critic will become more accurate to guide the policy change to a weighted variant of MI.
> > >
> > > Q3. "I doubt that for a general $w$ this notion of “weighted MI” would have the nice properties of mutual information"
> > >
> > >     A3: The existing studies on weighted MI is to study applications where certain commonality between objects or events are more significant than others. Since the weight in our case is the expected Q and value approximation, the weight amplifies the effect of MI when agents perform a strategy that is mutually deemed as an improvement. (i.e. avoid damage, long life time, etc). The question on whether it is symmetric, the MI formulation is to filter and learn the symmetric distribution (mutual), the distributions that are dissimilar will be ignored in the MI maximization optimization. This is of course not ideal, and why M&M performs a second stage of MI minimization to remedy the problem.
> > >
> > > Q4. & Q6 "... A.6... I don’t see how does it say anything about the convergence of $\theta$." & "The mutual information expression is the gradient of $J(\theta)$ objective."
> > >
> > >     A4: We believe Q4 and Q6 are related to why we need to take the "integral". Notice how Eq(3) and Eq(7) both have $\nabla$ and are in gradient form. Taking the integral will integrate away $\nabla$ and tell us what happen to a policy as it update with Eq(3)'s gradient over time. The effect of mutual information is from integrate many steps of such optimization until convergence. The accumulated gradient takes on the form the "long-term behavioral change" of MI maximization / minimization. In other words, applying 1 step of Eq(3) does not lead to MI maximization, but the whole optimization until convergence does.
> > >
> > > Q5. "Why the heuristic of pairing the agents and summing the gradients"
> > >
> > >     A5: The pairing of agents is essential to accelerate the learning of agents. When agents are paired with strong opponents, the policy optimization encourages agents to learn strategies to be robust against the possible exploits. The summing of the gradients is so that policy skill transfer can occur. (The two techniques are popularized by OpenAI Five and NeuPL for more efficient optimization)
> > >
> > > Q7. "BR strategy need not even converge" & "What does innate mean here?"
> > >
> > >     A7: In the BR variant proposed in NeuPL, the current joint policy learns a BR strategy profile against all past policies with weighted probability via graph solver. This is to address the rock-paper-scissors strategy cycle. Innate here means the unique character attributes of each agent. For an agent that can innately move faster than the other agents may benefit from learning strategies that utilize this advantage. Agents that are slow, but are built like a tank may specialize to a different set of policies. Our research here is to show that under joint policy optimization, MI maximization does  not  lead to such specialization for individuals. M&M is to first find the mutual joint policy, then form minimization on top to define specialization.
> > >
> > > We thank the Reviewer for diving deep and understand many critical aspects of our derivation. We hope we are able to address the above queries partially / fully. We appreciate the Reviewer, and look forward for more of these critical questions.
> > >
> > > (The revision period has just passed, please forgive us for unable to address the "Minor notation issue")

---

> > > > ### Comment · Reviewer_ryJg · 2022-11-29
> > > > **Still major issues remain.**
> > > >
> > > > 1. This question was more about the joint cumulative objective rather than whether we optimize the agent policies jointly or in a distributed manner. I could not find a precise mathematical definition of individual agent's objective. The authors directly started with an objective in an expected advantage form(in eqn 1) without clearly stating what those $Q, V$ functions are? Are they cumulative value of all the agents or are you modelling the value of each agent using a different $Q$?
> > > >
> > > > 2. So, the observation in section 2.3 that "policy gradient (Sutton et al., 1999) in multi-agent joint optimization can be reduced to MI maximization of an agent population." is not exactly correct. I also do not see any novel connection with the MI duality objective(mentioned in the title)  being used in the algorithm(the individual agents are simply trained to best respond to the current version of all other agents in line 8 of algorithm 2 in appendix). This raises a question on main contribution of the paper.
> > > >
> > > > 3,5,6,7 : In general, the answers provided do not look very convincing to me.
> > > >
> > > > 4. I see the first point but I do not see why in view of point 2(the $Q$'s keep changing), the convergence would take place? I feel the authors need to adapt theory to account for this actor-critic style update which would be too much change for this paper.
> > > >
> > > > Overall, even though the broader goal of this work is interesting, the contribution of paper in the current form is rather limited and the work has serious issues on some of the points that I have raised earlier, specifically the main claim on equivalence of policy gradient and MI maximization is too far fetched. So, at this point I could not recommend accepting this paper.

---

> > > > > ### Author Response · Authors · 2022-11-29
> > > > > **Respond**
> > > > >
> > > > > We thank the Reviewer for your time, suggestion and feedback.
> > > > >
> > > > > We will keep the feedback in mind and improve on the paper.
> > > > >
> > > > > Thank you.

---

> ### Author Response · Authors · 2022-11-11
> **Second Thread: Paper Revision #1 with Markdown**
>
> We have prepared a point-by-point response to the questions for each Reviewer in the first response thread. (Below this thread)
>
> For our first paper revision, we have uploaded a red-line / boxed highlight revision of our paper.
>
>       The red-lines address the specific issues pointed out by the individual reviewers.
>
>       The boxed highlighter highlights portions of paper that are added to better clarify, define the undefined notations, terms, definitions of social graph and more.
>
>
> Here is what we have done on the paper revision:
>
>       1. We added introductory paragraphs to problem formulation, precise notations and functions definitions to improve - writing clarity
>
>       2. A couple of paragraphs (i.e. section 2.3, 3) have been rewritten with clean descriptions of the proposed methodology to improve - writing clarity.
>
>       3. Generalists, Specialists, social graph definitions and descriptions have been added to the paper explicitly.
>
>       4. We moved our pseudocode into the appendix for extra space in the main text.
>
>       5. We added additional details on Experiment 4.5’s analytic discussion.
>
>       6. We have extended Conclusion to summarize the paper.
>
> We are thankful for the invaluable feedback from the reviewers.
> The paper clarity has notably improved because of our discussion.
>
> For the following week, we appreciate your time and would like to seek additional feedback. We will continue to improve the paper quality until the paper is satisfactory to you.
> Thank you!

---

### Author Response · Authors · 2022-11-11
**Round#1 Paper Feedback Summary:**

Dear Reviewer,

Thank you for your time and review feedback.
Here we summarized the common feedback pulled from the 8 paper reviews:

Strength:

    1. “Paper is well motivated”

    2. “The paper is backed by a succinct theoretical analysis”

    3. “The paper considers an interesting problem which is relevant to the MARL community.”

    4. “Novelty: In my perspective, the approach has a certain novelty in it.”

Weakness:

    1. Poor writing clarity and typos

    2. Undefined notations, terms, claim of novelty

    3. Should formal define Generalists and Specialists

    4. Should explicitly define social graph and graph solver

Interesting sections to discuss more:

    Elaborate the discussion on experiment 4.5 on the inconsistency of agents’ performance gain.

---

> ### Author Response · Authors · 2022-11-18
> **Paper Revision Links and Descriptions**
>
> Revision #1:
>
> This is our paper's first revision with red lines that mainly to address undefined notations, terms and improve overall paper clarity.
>
>       https://openreview.net/references/pdf?id=pFfR2GWRT
>
> Revision #2:
>
> This is our paper's second revision. We have done a minor revision on:
>
>       https://openreview.net/references/pdf?id=HKkPWyzEQ
>
> 1. More precisely defined \epsilon-Nash (section 3.1)
>
> 2. Added short outline of experiment (section 4)
>
> 3. Fig. 4 to 2 bars per agents (section 4.5)
>
> 4. Revised caption and discussion for Fig. 5 (section 4.6)
>
> So far we have received "Good Improvement" on the clarity of the paper as feedback.
>
> We thank all the Reviewers for your support and constructive feedback. Thank you!

---

### Decision · Program_Chairs · 2023-01-20

**Decision:**

Reject

**Justification For Why Not Higher Score:**

The reviewers pointed out several weaknesses in the paper and reached a rejection decision.


**Justification For Why Not Lower Score:**

N/A

**Metareview: Summary, Strengths And Weaknesses:**

The reviewers agreed that the paper proposes a novel minimax formulation of mutual information for multi-agent policy optimization. However, the reviewers pointed out several weaknesses in the paper and shared common concerns. We want to thank the authors for their detailed responses. Based on the raised concerns and follow-up discussions, unfortunately, the final decision is a rejection. Nevertheless, the reviewers have provided detailed and constructive feedback. We hope the authors can incorporate this feedback when preparing future revisions of the paper.